# Enhancing operational stability of OLEDs based on subatomic modified thermally activated delayed fluorescence compounds

Sinyeong Jung[1,2,6], Wai-Lung Cheung [1,6], Si-jie Li[1,6], Min Wang[1], Wansi Li[1], Cangyu Wang[1], Xiaoge Song[1], Guodan Wei [1,2] ✉, Qinghua Song [1], Season Si Chen [3] ✉, Wanqing Cai[4], Maggie Ng [1], Wai Kit Tang[5] & Man-Chung Tang [1] ✉

The realization of operationally stable blue organic light-emitting diodes is a challenging issue across the field. While device optimization has been a focus to effectively prolong device lifetime, strategies based on molecular engineering of chemical structures, particularly at the subatomic level, remains little. Herein, we explore the effect of targeted deuteration on donor and/or acceptor units of thermally activated delayed fluorescence emitters and investigate the structure-property relationship between intrinsic molecular stability, based on isotopic effect, and device operational stability. We show that the deuteration of the acceptor unit is critical to enhance the photostability of thermally activated delayed fluorescence compounds and hence device lifetime in addition to that of the donor units, which is commonly neglected due to the limited availability and synthetic complexity of deuterated acceptors. Based on these isotopic analogues, we observe a gradual increase in the device operational stability and achieve the long-lifetime time to 90% of the initial luminance of 23.4 h at the luminance of 1000 cd m$^{-2}$ for thermally activated delayed fluorescence-sensitized organic light-emitting diodes. We anticipate our strategic deuteration approach provides insights and demonstrates the importance on structural modification materials at a subatomic level towards prolonging the device operational stability.

To realize blue organic light-emitting diodes (OLEDs) with high efficiency and long operational lifetime, it is important to suppress bimolecular triplet-triplet annihilation (TTA) or and triplet-polaron annihilation (TPA) which could induce undesirable exciton fusion processes such as high-energy polaron formation[1,2]. For example, the employment of a stacked device structure with two emissive layers (EMLs) with graded dopant concentrations[3], the introduction of an

excited state manager molecule as a sacrificing agent[4], and the use of sky-blue thermally activated delayed fluorescence (TADF) emitters as assistant dopants[5] have been shown to effectively lengthen the device lifetime. While intensive efforts have been made in terms of device engineering[6], there has been a recent interest on molecular engineering to investigate the structure-property relationship with strategies that pinpoint on the emitter or host material design from a

[1]Institute of Materials Research, Tsinghua Shenzhen International Graduate School, Tsinghua University, 518055 Shenzhen, China. [2]Tsinghua-Berkeley Shenzhen Institute (TBSI), Tsinghua University, 518055 Shenzhen, China. [3]Institute of Environment and Ecology, Tsinghua Shenzhen International Graduate School, Tsinghua University, 518005 Shenzhen, China. [4]Faculty of Materials Science, MSU-BIT University, 518172 Shenzhen, China. [5]Department of Chemistry, Faculty of Science, University of Malaya, 50603 Kuala Lumpur, Malaysia. [6]These authors contributed equally: Sinyeong Jung, Wai-Lung Cheung, Si-jie Li ✉e-mail: weiguodan@sz.tsinghua.edu.cn; season.chen@sz.tsinghua.edu.cn; kobetang2021@sz.tsinghua.edu.cn

chemical point of view[7,8], which are of equal importance to enhance the device operational lifetime. Specifically, a particular focus has been put on the exploration of factors such as the bond dissociation energy (BDE) of host materials[9] and deuterium substitution of host materials[10–12], phosphorescent (Ph) emitters[13–18] or donor units of TADF emitters[19,20]. These have prompted our interest to take a deeper investigation on the isotopic effect induced by different components within the material on the device operational stability.

The kinetic isotopic effect (KIE) is a mechanistic phenomenon associated with one of the atoms being replaced by its isotope with the observation of a change of rate of a chemical reaction[21]. Among various isotopic effect, the replacement of proton, or protium, by deuterium is the most common strategy considering the relative ease and cost of synthesis, as well as the more prominent effect based on the change of mass, compared to other isotope effects such as carbon-13[22]. For hydrogen isotopic effect, it is often expressed by the ratio of $k_H/k_D$ in the range of 1–7 depending on the site of substitution, where primary hydrogen KIE refers to the substitution at the bond breaking site and secondary hydrogen KIE refers to the substitution other than the bond breaking site in the rate determining step. According to Eq. (1),

$$v = \frac{1}{2\pi}\frac{\sqrt{k}}{\sqrt{u}} \tag{1}$$

where $v$ is the vibration frequency, $k$ is the force constant, and $u$ is the reduced mass, the substitution of an atom by its heavier isotope requires a higher activation energy for bond dissociation. Therefore, it is anticipated that deuterium substitution could enhance the intrinsic stability of a molecule. This has been evidenced by a lower vibrational zero point energy (ZPE) of deuterated polymers[23] and an increased photoluminescence quantum yield (PLQY) of near-infrared emitters owing to the lower non-radiative decay rate ($k_{nr}$) through a reduced reorganization energy ($\lambda$) and exciton-vibration coupling (i.e. a smaller Franck–Condon factor ($F$))[17]. To date, only few research has been reported to study the improved molecular stability through deuterium substitution in OLED applications[11,12,20]. For instance, Adachi and co-workers reported a host material with deuterated carbazole donors

that exhibited a 2-fold enhancement of device half lifetime, i.e. $LT_{50}$ at an initial luminance of 1000 cd m$^{-2}$, from 17 h to 40 h compared to its non-deuterated counterpart[11]; while the isotopic effect has been shown positive on the thermal stability and electroluminescence (EL) of TADF emitters[20], the underlying mechanism of such effect has not yet been studied in correlation with the corresponding device operational stability. More importantly, to the best of our knowledge, studies on deuterated single emitting materials are solely based on donor units of TADF compounds[19,20,24], or random deuteration of host materials[12], suggesting the capacity of fully deuterated materials, as well as the position and degree of deuteration of different components of a molecule with respect to the distances to potential bond breaking sites, e.g. C−N bond[25,26], has yet to be unlocked.

In light of these, our interest focuses on the investigation of the isotopic effect through independent deuteration of the donor and acceptor units. We take advantage of the symmetrical, robust and rigid oxygen-bridged triarylboron unit[27,28] to study such effect in a series of deuterated isotopic analogues, with a targeted deuteration of diphenylamine (DPA) and/or phenoxy groups on the 5,9-dioxa-13b-boranaphtho[3,2,1-de]anthracene (BO) moieties, namely DPA-BO, d-DPA-BO, DPA-d-BO and d-DPA-d-BO (Fig. 1a). We conduct a comprehensive study on the deuterated effect towards their photophysical behavior, intrinsic stability and device operational lifetime. Under the irradiation of a high-power lamp, we found that the photostability of the compounds increases with the degree of deuteration as observed from the time-dependent UV-vis and PL spectra in solution, neat film and doped film, supported by theoretical studies where the increased BDE of the C − D bond imposes an additive effect on the molecules. These results suggest the important role of deuterated acceptor unit that is critical to enhance the overall stability of the compounds, which is commonly neglected due to their limited commercial availability and synthetic difficulty. Based on this series of compounds, we observe a gradual increase in the device operational stability and achieve the $LT_{90}$ of 23.4 h at the luminance of 1000 cd m$^{-2}$ for TADF-sensitized OLEDs. We hope that the present study provides insights into further boosting the device operational stability through the strategic deuteration of luminescent compounds at the subatomic level.

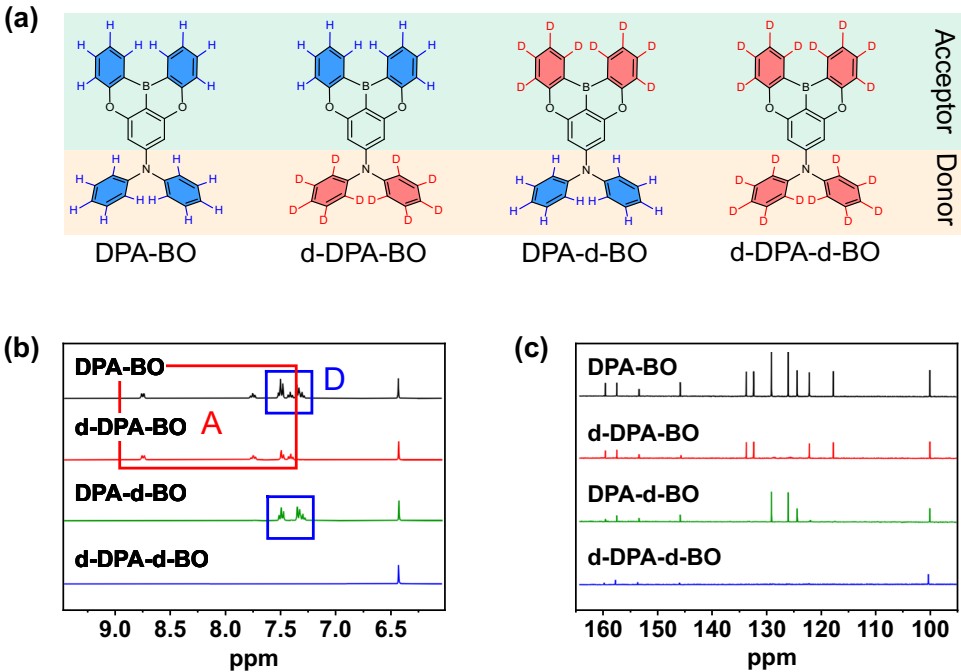

**Fig. 1 | Molecular structures and characterization of synthesized compounds. a** Chemical structures of the investigated TADF emitters. **b** $^1$H NMR stacked spectra. **c** $^{13}$C NMR stacked spectra.

## Results

### Synthesis and characterization

Supplementary Fig. 1–3 illustrate the synthetic route for precursor compounds BO, d-BO, and d-DPA. The series of deuterated isotopic analogues was synthesized by Buchwald–Hartwig coupling reaction, as shown in Supplementary Fig. 4, and well-characterized by [1]H and [13]C{[1]H} nuclear magnetic resonance spectroscopy and high-resolution electrospray ionization mass spectrometry (Supplementary Fig. 5–21). As expected, the proton signals of d-BO and d-DPA moieties are not found in respective compounds, while a singlet peak centered at 6.52 ppm exists in all compounds which belongs to the protons on the central aromatic ring (Fig. 1b). In addition, we have also observed the peaks corresponding to the carbon atom bonded with deuterium showing a weaker signal in the [13]C-NMR spectra (Fig. 1c), possibly attributed to the different magnetic moment of proton and deuterium. Meanwhile, we observe a concomitantly drop in the absorption peak at ca. 3040 cm$^{-1}$ and emergence of the peak at ca. 2280 cm$^{-1}$ in the infrared (IR) spectra upon increasing the degree of deuteration, corresponding to the C−H and C−D bond sketches, respectively (Supplementary Fig. 22a). These results are in good agreement with the simulated IR spectra of the compounds (Supplementary Fig. 22b). The TADF compounds also show high decomposition ($T_d$) and glass transition ($T_g$) temperatures of over 340 °C and 240 °C, respectively (Supplementary Fig. 23 and Supplementary Table 1), proving to be thermally stable enough to be applied as emissive dopants in vacuum-deposited OLEDs.

### Theoretical investigation

To gain insights into the nature of the absorption and emission origins, the excited states involved in the TADF process, density functional theory (DFT) and time-dependent density functional theory (TDDFT) calculations were performed on these compounds. The spatial plots of the highest occupied molecular orbital (HOMO) and lowest unoccupied molecular orbital (LUMO) of representative compound DPA-BO at the optimized ground-state geometry are shown in Fig. 2, while those of other selected molecular orbitals are illustrated in Supplementary Fig. 24–27. Apparently, the HOMOs of these compounds are predominantly the π-orbitals localized on the DPA moiety, while the LUMOs are predominantly the π* orbitals localized on the BO moiety. Based on the simulated absorption spectra of all compounds, as shown in Supplementary Fig. 28, and their first fifteen singlet excited states listed in Supplementary Table 2, we assign the low-energy absorption band to be originating from the S$_1$ state with charge-transfer (CT) character from the DPA moiety to BO moiety. The emission energies of the S$_1$ state are estimated to be ca. 408 nm (Supplementary Table 3). Supplementary Fig. 29–31 illustrate the selected molecular orbitals of DPA-BO at the optimized S$_1$, T$_1$, and T$_2$ states geometry. Apparently, the S$_1$ and T$_1$ states are of the CT character ($^1$CT and $^3$CT, respectively) while the T$_2$ state is of the localized state character ($^3$LE). The computed energy differences between the S$_1$ and T$_1$ states, i.e. $\Delta E(S_1-T_1)$, and that between the S$_1$ and T$_2$ states, i.e. $\Delta E(T_2-S_1)$, are found to be 0.22 eV and 0.08 eV, respectively. Meanwhile, the computed values of the spin-orbit coupling (SOC) matrix elements[29], $\langle^1CT|\hat{H}_{SOC}|^3CT\rangle$ and $\langle^1CT|\hat{H}_{SOC}|^3LE\rangle$, are found to be 0.150 cm$^{-1}$ and 0.103 cm$^{-1}$, respectively. Although heavy atom could improve the SOC, the same estimated SOC constants for all compounds also imply an insignificant impact on the SOC upon deuteration of the compounds[20].

To estimate the effect of deuteration, we compute the ZPE, and the BDE of C−H (or C−D) bond at ground-state geometry as well as C−N bond between donor and acceptor units, of the compounds at the S$_0$ and T$_1$ states. The ZPE of these compounds and DPA and BO precursor compounds are listed in Supplementary Table 4, whereas the BDE of the C−N bond and the C−H (or C−D) bond are summarized in Supplementary Tables 5, 6. Regarding the C–N bond dissociation, the ZPE decreases from 262.9 kcal mol$^{-1}$ to 226.1 kcal mol$^{-1}$ upon increasing the degree of deuteration in general (Supplementary Table 4). It should be

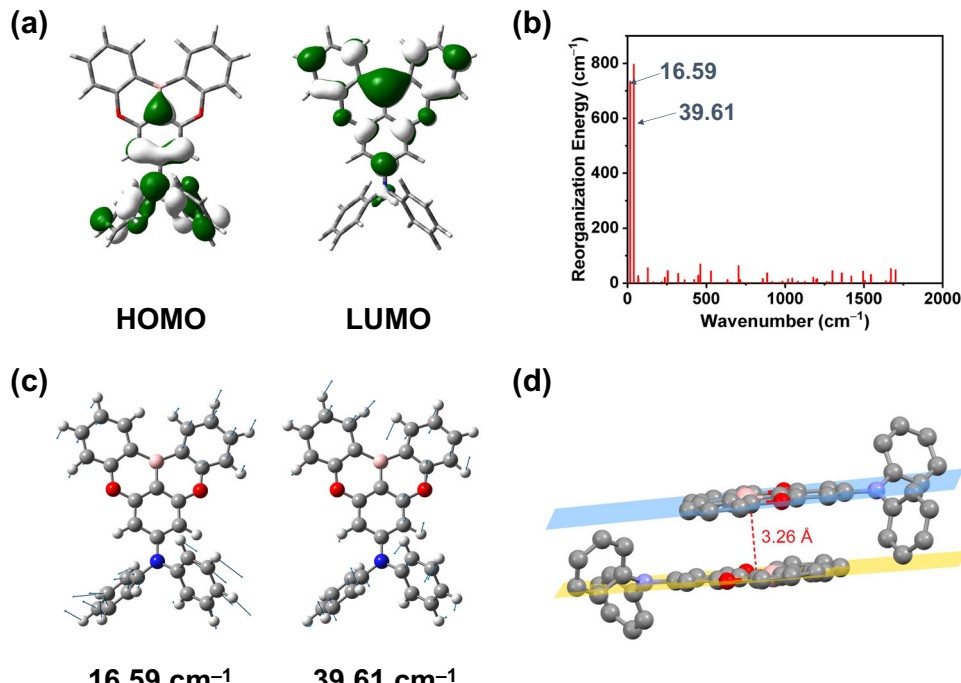

**Fig. 2 | Theoretical investigation of the deuterated isotopologues. a** Spatial plots (isovalue = 0.03) of the HOMO and LUMO for the selective compound DPA-BO at the optimized ground-state geometry. **b** Plots of computed reorganization energies as a function of normal mode wavenumbers for S$_1$-S$_0$ of DPA-BO.

**c** Illustration of selected normal modes contributing to large reorganization energies for the S$_0$ state of DPA-BO. **d** Ground-state geometry of the dimer of DPA-BO optimized at the M06 level.

noted that the addition of ZPE values of DPA and BO precursor compounds are not equal to the ZPE of the final compounds. The BDE of the C–N bond at the $S_0$ state is found to drop slightly upon deuterating the compounds, whereas the values at the $T_1$ state do not show apparent changes and are ca. 0.66 eV. The transition states of the C–N bond dissociation of the $T_1$ state of DPA-BO and d-DPA-d-BO have been located and their enthalpies of activation are summarized in Supplementary Table 7. The enthalpies of activation are ca. 0.90 eV and they show insignificant difference upon deuteration. Therefore, from both the thermodynamic and kinetic aspects, deuteration of DPA-BO does not strengthen the C–N bond, nor does it influence the rate of C–N bond dissociation. On the other hand, the BDE of the C–D bond is higher than that of the C–H bond by ca. 0.1 eV. With multiple C–D bonds existing in the compounds (i.e. 10 in d-DPA-BO, 8 in DPA-d-BO; 18 in d-DPA-d-BO), the additive effect should be appreciable, and hence the proton abstraction reaction is thermodynamically less favorable and subsequent formation of high molecular adducts is probably suppressed[12,30,31]. In short, upon deuteration of DPA-BO, though the C–N bond is slightly weakened, the strengthening of multiple C–D bonds in the compounds is suggested to prevent most of the proton abstraction reaction and subsequent radical formations. With these calculated results, we anticipate that the higher degree of deuteration on both donor and acceptor units should exert a more prominent effect on the device stability.

In addition, the effects of deuteration of the compounds on their Huang-Rhys factor ($S_j$), reorganization energy ($\lambda_j$), radiative ($k_r$) and nonradiative ($k_{nr}$) decay rate constants have been investigated. The plot of the computed $\lambda_j$ as a function of normal mode wavenumbers of DPA-BO for $S_1$-$S_0$ is shown in Fig. 2b, whereas those of other molecules are shown in Supplementary Fig. 32. It is obvious that the $\lambda_j$ is contributed predominantly by the low-frequency vibrational modes under 50 cm$^{-1}$. The low-frequency vibrational modes of DPA-BO are illustrated in Fig. 2c and those of other molecules are shown in Supplementary Fig. 33 and 34. They are mainly the rotation of the phenyl rings in both the DPA and BO moieties, which contribute predominantly to the distortion of the $S_1$ geometry from the $S_0$ geometry. The displacement ($\Delta Q$), Huang–Rhys factors ($S_j$) and reorganization energies of selected normal modes of the compounds are summarized in Supplementary Tables 8, 9, and the computed $k_r$ and $k_{nr}$ values are summarized in Supplementary Table 10. There is no obvious trend in $\Delta Q$, $S_j$ and $\lambda_j$ upon the deuteration of the compounds, and the $k_r$ ($1.18 \times 10^8\,s^{-1}$) and $k_{nr}$ (ca. $1.4 \times 10^{11}\,s^{-1}$) are nearly the same upon deuteration. Therefore, deuteration does not have significant effect on $k_r$ and $k_{nr}$ in this case, resemblance to our photophysical studies.

To further investigate the possible intermolecular interactions in the solid-state thin film, DFT calculation based on the dimer form of DPA-BO has been performed at the M06 level of theory. The optimized geometry of the dimer of DPA-BO is shown in Fig. 2d. The two planes of the BO moieties are nearly parallel and they are separated with a distance of 3.26 Å, which is within the separation between the π–π planes (ca. 3.35 Å), indicating the presence of π–π interactions in the solid-state thin film.

## Photophysical and TADF properties

To well-characterize the electronic absorption, emissive and TADF properties of the deuterated isotopic analogues, we conduct steady-state UV-vis absorption and emission spectroscopy to study the photophysical behavior in both solution and solid-state thin film. Figure 3a shows the absorption and emission spectra of the compounds in degassed toluene and Supplementary Table 11 summarizes their photophysical data. In general, all compounds show moderate absorption bands at ca. 280−320 nm and intense absorption bands at 375 nm, attributing to the spin-allowed intraligand (IL) π→π* transitions of the BO and DPA moieties and ligand-to-ligand charge transfer

(LLCT) [π(DPA)→π*(BO)] transitions, respectively[28]. Upon excitation at λ ≥ 340 nm, all compounds feature a broad structureless fluorescence emission band with emission maxima peaking at 430 nm with a decay lifetime of ca. 4.4 ns, which is assigned as originating from the LLCT $^1$[π(DPA)→π*(BO)] excited state. The effect of deuteration is found to be insignificant towards the photophysical properties of the TADF compounds, in agreement with the theoretical results. We also look into the solvent-dependent emission behavior considering the CT nature of the excited state of these compounds. Upon increasing the solvent polarity from cyclohexane to dichloromethane, we observe a bathochromic shift of the LLCT band maxima from ca. 400 nm to 470 nm (Supplementary Fig. 35), confirming the strong CT character of the emissive state.

Next, we study the photophysical and TADF properties of the compounds doped in bis[2-(diphenylphosphino)phenyl]ether oxide (DPEPO) thin films. Table 1 summarizes the TADF characteristics of the investigated compounds. The photoluminescence (PL) spectra recorded at room temperature for DPA-BO at 5−20 wt% are shown in Supplementary Fig. 36. It is clearly seen that the structureless blue emission band exhibits a bathochromic shift upon increasing the dopant concentration, suggesting a concentration-dependent behavior arising from π−π stacking of the BO moiety[32,33]. This could be evidenced by the 3D images generated from time-of-flight secondary ion mass spectrometry (ToF-SIMS) which give information on the spatial localization of the host and emitting materials. Figure 3b shows the ToF-SIMS characteristics of the representative 10 wt% DPA-BO in DPEPO and those of the others are shown in Supplementary Fig. 37. In these images, we directly observe that the TADF compounds are not well-dispersed in the host matrix, indicating the presence of strong intermolecular interactions in the solid state, consistent with our computational results, which could possibly enhance the CT characteristics of the compounds[34]. On the other hand, the PL transient emission spectra recorded at low temperature and the steady-state emission spectra at room temperature, representing the phosphorescence and fluorescence emission of the compounds doped at 10 wt% in DPEPO (Supplementary Fig. 38), where an estimated $\Delta E(S_1-T_1)$ of ca. 0.08 eV is found for all compounds. As shown in Fig. 3c, the transient PL decay curves of these compounds can be well-fitted by the biexponential model, in which the prompt and delayed lifetimes are found to be ca. 3.5 ns and 59−67 μs, respectively. We also analyze the temperature-dependency of $k_{RISC}$ in 10 wt% of the compounds in DPEPO. Based on the Arrhenius Eq. (2):

$$k_{RISC} = A\exp(-E_a/k_{BT}) \qquad (2)$$

where $A$ is the frequency factor involving the SOC constant, $E_a$ is the activation energy, and $k_{BT}$ is the Boltzmann distribution, the estimated $E_a$ values are in the range of 37−41 meV (Supplementary Fig. 39), much smaller than the activation energy needed for reverse intersystem crossing (RISC) at 300 K ($k_B T \approx 25.9$ meV). The HOMO and LUMO energies of these compounds, estimated by ultraviolet photoelectron spectroscopy (UPS) (Supplementary Fig. 40) and UV−vis spectroscopy, are also listed in Supplementary Table 11. These experimental results are generally in line with the theoretical studies. Based on the above theoretical and experimental results, we deduce that the small $\Delta E(S_1-T_2)$, together with the ineligible SOC constant[20], suggest possible RISC process from the $T_2$ to $S_1$ states[35,36]. The schematic diagram of the energy levels and SOC matrix elements between the $S_1$ and $T_n$ states are shown in Fig. 3d.

## Intrinsic photostability

To investigate the effect of deuteration on the intrinsic photostability of the TADF compounds, we perform PL stability test on the compounds under a continuous irradiation with a CW ozone-free xenon arc lamp at room temperature. Supplementary Fig. 41 and 42 show

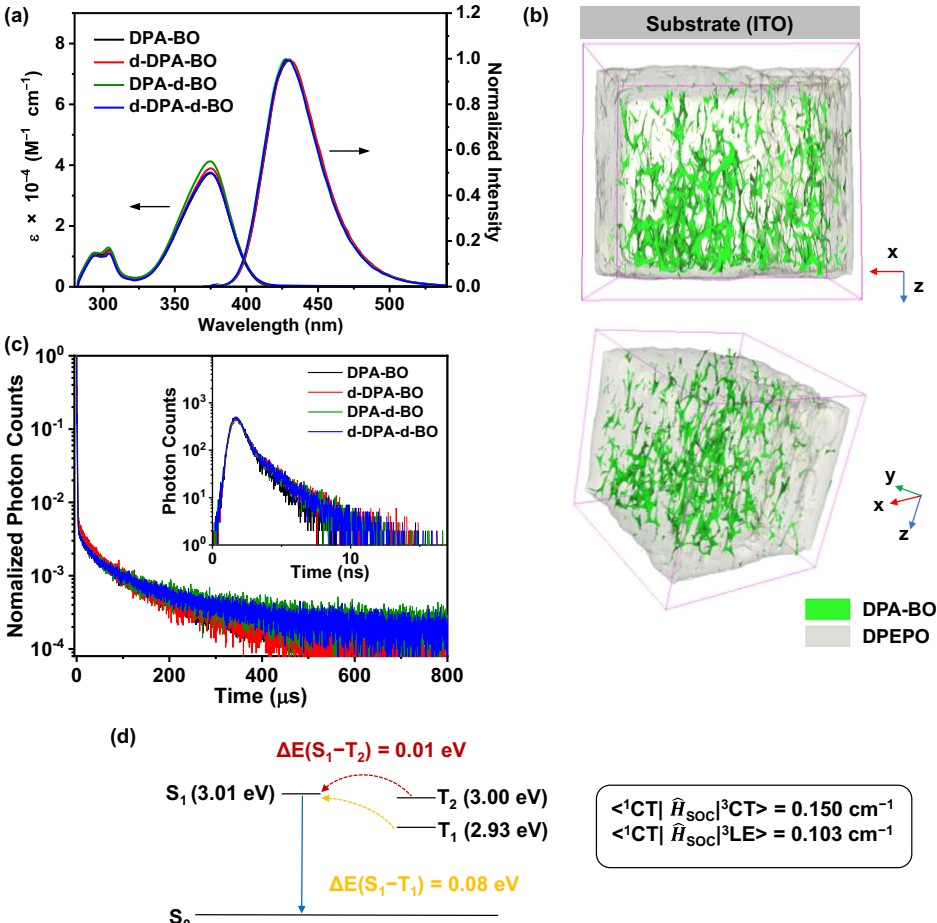

**Fig. 3 | Photophysical properties and spatial localization in DPEPO film.**
**a** UV–Vis and PL spectra recorded in toluene ($2 \times 10^{-5}$ M). **b** Spatial localization of 10 wt% DPA-BO ($C_{30}H_{20}BNO_2^+$, molar mass = 437.16 g/mol) doped in DPEPO ($C_{36}H_{28}O_3P_2^+$, molar mass = 570.09 g/mol). **c** Prompt (inset) and delayed PL decay of 10 wt% doped films. **d** Schematic diagram of the energy levels and SOC matrix elements between the $S_1$ and $T_n$ states.

**Table 1 | TADF characteristics of the investigated compounds**

| Compound | $\Phi_{PL}$ (%)[a] | $\Phi_p$ (%)[b] | $\Phi_d$ (%) | $\tau_p$ (ns)/$\tau_d$ (μs)[c] | $k_{pf}$ ($10^7$ s$^{-1}$)[d] | $k_{df}$ ($10^4$ s$^{-1}$)[e] | $k_{ISC}$ ($10^7$ s$^{-1}$)[f] | $k_{RISC}$ ($10^4$ s$^{-1}$)[g] |
|---|---|---|---|---|---|---|---|---|
| DPA-BO | 89.0 | 68.9 | 20.0 | 3.5/67.2 | 19.6 | 0.3 | 4.4 | 0.4 |
| d-DPA-BO | 88.3 | 67.1 | 21.1 | 3.4/56.8 | 19.2 | 0.3 | 4.6 | 0.4 |
| DPA-d-BO | 89.3 | 63.9 | 25.4 | 3.4/64.1 | 18.7 | 0.4 | 5.3 | 0.6 |
| d-DPA-d-BO | 90.2 | 68.8 | 21.4 | 3.4/59.5 | 20.1 | 0.4 | 4.8 | 0.5 |

[a]Absolute PLQYs evaluated under $N_2$ of 10 wt% DPEPO thin film (298 K).
[b]Absolute PLQYs evaluated under air of 10 wt% DPEPO thin film (298 K).
[c]PL lifetimes of prompt ($\tau_p$) and delayed ($\tau_d$) decay components in 10 wt% DPEPO thin film (298 K).
[d]$k_{pf}$, prompt fluorescence rate constant[59].
[e]$k_{df}$, delayed fluorescence rate constant[59].
[f]$k_{ISC}$, intersystem crossing rate constant[59].
[g]$k_{RISC}$, reverse intersystem crossing rate constant[59].

the UV-vis and PL spectra of the compounds in degassed toluene solution. Apparently, the shape of the absorption and emission bands remains unchanged, but the percentage of the CT absorption band at ca. 370 nm and the CT emission band at ca. 440 nm generally decreases to a lesser extent upon increasing the degree of deuteration, i.e., from 70.6% to 54.7% in the UV-vis spectra (Figs. 4a and 4b), and from 91.4% to 81.9% in the PL spectra, indicating a positive deuteration effect on the photostability of the compounds[12]. Similar observations are found for the compounds doped in DPEPO film (Supplementary Fig. 43). To confirm the CT character in the absorption and emission bands, we perform in-situ EPR experiments and observe the generation of *N*-radicals in the TADF compounds in

powder form. Supplementary Fig. 44 shows the time-dependent EPR spectra of all compounds, where we observe the accumulation of long-lived radicals during irradiation, and a single spin signal with *g* value at 2.0030 that is consistent with the reported aminopyridine radical[37,38]. To better understand the effect of deuterated emitters to the photostability under a host-guest environment, we also perform PL stability test of the TADF compounds doped in mCBP and the PL intensity drop over time is shown in Figs. 4c and 4d. Upon increasing the degree of deuteration, the percentage of the CT emission band drops from 69.1% to 56.4%, further confirming the important role of deuterated acceptors in addition to deuterated donors in the TADF compounds.

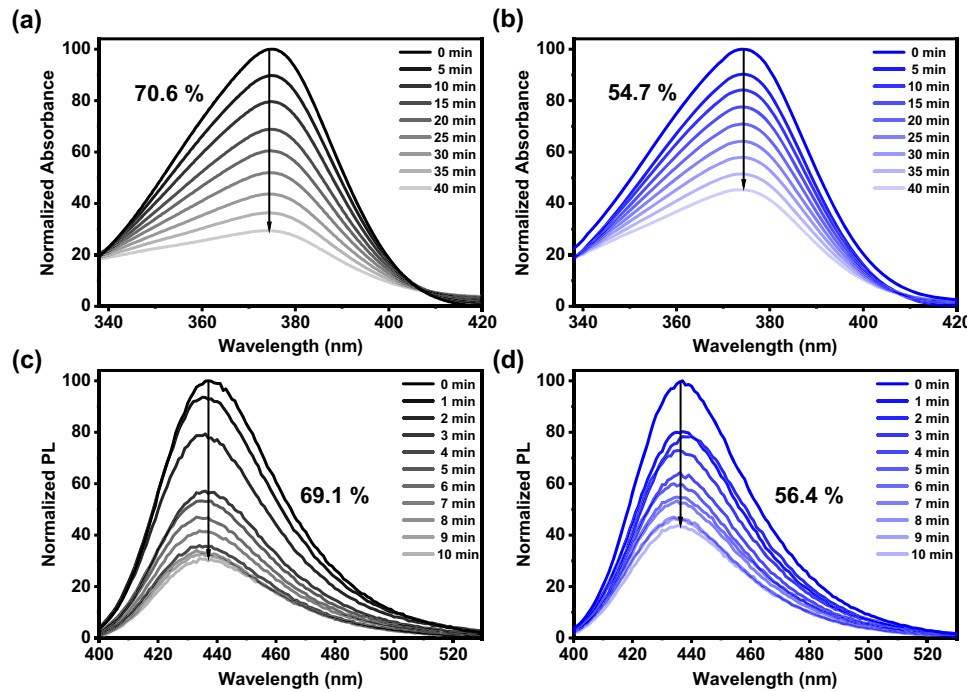

**Fig. 4 | Photolysis results.** UV–Vis spectra of (**a**) DPA-BO, (**b**) d-DPA-d-BO at $2 \times 10^{-5}$ M in toluene solution over 40 min and PL spectra of 10 wt%, (**c**) DPA-BO, (**d**) d-DPA-d-BO doped in mCBP over 10 min under irradiation of 300 W xenon lamp.

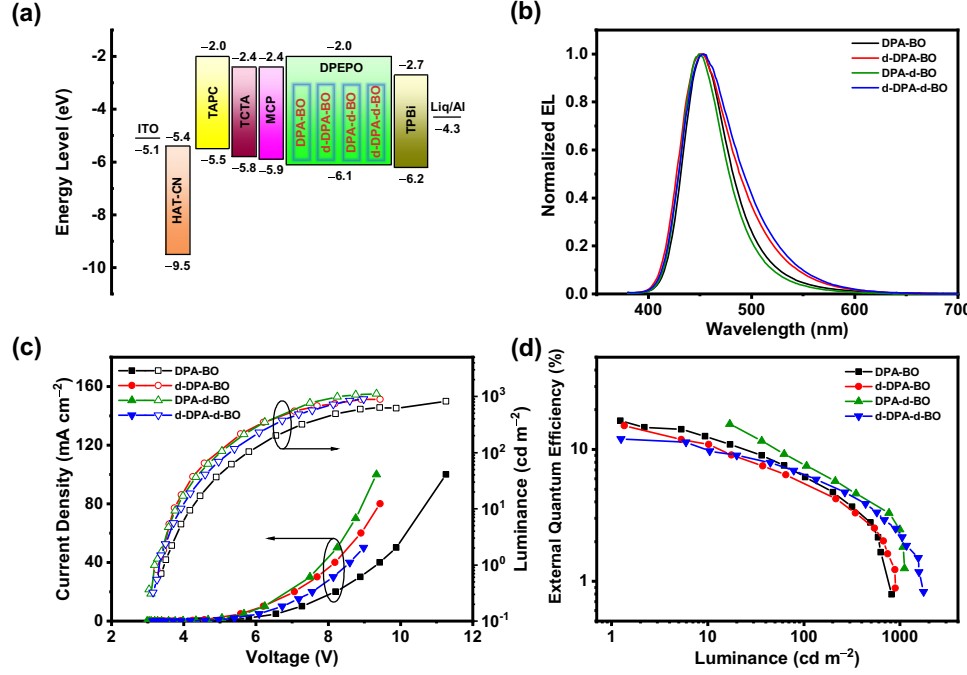

**Fig. 5 | Device characteristics of the vacuum-deposited OLEDs based on 15 wt% of the compounds doped in DPEPO. a** Device configuration and energy-level diagram. **b** Normalized EL spectra. **c** $J$−$V$−$L$ plots. **d** EQE versus luminance.

## Device characterization and correlation of intrinsic and operational stability

To confirm the TADF properties of the compounds, we fabricate a series of vacuum-deposited devices based on the dopant concentration of 5−20 wt% in DPEPO exhibiting blue EL. Figure 5 shows the device characteristics of 15 wt% of the compounds doped in DPEPO, while those of the others are shown in Supplementary Fig. 45 and summarized in Table 2. Apparently, these devices exhibit a blue EL

emission peak ($\lambda_{EL}$) at ca. 450 nm which corresponds to the CIE chromaticity coordinates of (0.15, 0.10). The high resemblance of the EL spectra to the corresponding PL spectra indicates a well confinement of excitons in the DPEPO host. As seen from the $J$−$V$−$L$ device characteristics, the devices exhibit maximum EQEs of over 17%, further confirming the TADF properties of these compounds. Unfortunately, these devices exhibit limited operational lifetime, probably due to the close proximity of TADF molecules that lead to aggregated species and

**Table 2 | Device characteristics of the vacuum-deposited OLEDs based on the TADF compounds doped in DPEPO**

| Compound | Conc. (wt%) | $V_{on}$[a] (V) | $L_{max}$[b] (cd m$^{-2}$) | $\lambda_{max}$[c] (nm) | CIE[d] (x, y) | EQE$_{max/100/1000}$[e] (%) | $\Delta_{roll\text{-}off}$[f] (%) | CE[g] (cd A$^{-1}$) | PE[h] (lm W$^{-1}$) |
|---|---|---|---|---|---|---|---|---|---|
| DPA-BO | 5 | 4.4 | 106 | 443 | 0.15, 0.08 | 11.6/1.3/- | 88 | 5.3 | 3.7 |
| | 10 | 3.7 | 282 | 449 | 0.15, 0.09 | 13.4/4.1/- | 70 | 8.0 | 5.9 |
| | 15 | 3.5 | 629 | 453 | 0.14, 0.11 | 16.5/6.2/- | 54 | 12.4 | 11.1 |
| | 20 | 3.3 | 1361 | 454 | 0.14, 0.12 | 17.1/7.0/1.7 | 60 | 14.6 | 13.5 |
| d-DPA-BO | 5 | 4.1 | 138 | 439 | 0.16, 0.08 | 13.1/1.9/- | 85 | 7.3 | 5.5 |
| | 10 | 3.6 | 342 | 446 | 0.15, 0.09 | 15.5/3.7/- | 74 | 11.1 | 9.8 |
| | 15 | 3.3 | 890 | 450 | 0.15, 0.10 | 15.1/5.6/- | 63 | 13.6 | 12.7 |
| | 20 | 3.2 | 1511 | 454 | 0.14, 0.11 | 14.6/6.1/2.1 | 59 | 14.0 | 13.7 |
| DPA-d-BO | 5 | 3.9 | 153 | 443 | 0.15, 0.08 | 12.4/3.2/- | 76 | 4.2 | 2.8 |
| | 10 | 3.4 | 451 | 449 | 0.14, 0.09 | 14.1/5.9/- | 60 | 6.8 | 5.0 |
| | 15 | 3.2 | 1111 | 450 | 0.14, 0.10 | 15.6/7.8/2.4 | 50 | 9.3 | 8.2 |
| | 20 | 3.2 | 1672 | 453 | 0.14, 0.11 | 14.8/8.6/3.5 | 42 | 9.4 | 7.6 |
| d-DPA-d-BO | 5 | 4.0 | 156 | 443 | 0.16, 0.07 | 13.5/2.1/- | 85 | 7.2 | 5.4 |
| | 10 | 3.6 | 456 | 446 | 0.15, 0.09 | 15.4/4.3/- | 77 | 11.3 | 9.9 |
| | 15 | 3.3 | 895 | 450 | 0.15, 0.10 | 16.3/6.0/- | 64 | 14.8 | 13.8 |
| | 20 | 3.2 | 1758 | 454 | 0.15, 0.11 | 12.6/6.5/2.3 | 49 | 12.6 | 12.1 |

[a]$V_{on}$ represents turn-on voltage.
[b]$L_{max}$ represents maximum luminance.
[c]$\lambda_{max}$ represents peak maximum.
[d]CIE coordinates are taken at luminance of 100 cd m$^{-2}$.
[e]EQE$_{max}$ represents maximum external quantum efficiency.
[f]Efficiency roll-off measured by $\Delta_{roll\text{-}off} = 1 - (EQE_{100}/EQE_{max})$.
[g]CE represents maximum current efficiency.
[h]PE represents maximum power efficiency.

the relatively long excited-state lifetime in solid-state thin film, as evidenced above. In this regard, we turn to fabricate TADF-sensitized OLEDs to study the relationship between the molecular intrinsic stability, based on isotopic effect, and device operational stability. The device characteristics of vacuum-deposited TADF-sensitized OLEDs with t-Bu-$v$-DABNA[39] or BN3[40] as the emitter are illustrated in Fig. 6 and summarized in Table 3. These devices exhibit sharp EL bands peaking at ca. 461 nm or 572 nm, respectively, with maximum EQEs of over 20%. Notably, the device operational stability is found to follow the trend upon increasing the degree of deuteration of the TADF compounds, in which the LT$_{50}$ of the devices are found to be increased from ca. 20 h to 27 h at the luminance of 100 cd m$^{-2}$ for blue-emitting devices with CIE coordinates of (0.13, 0.09). It is worth mentioning that we observe a gradual increase in the LT$_{90}$ of the devices from ca. 12 h (DPA-BO), ca. 15 h (d-DPA-BO), ca. 19 h (DPA-d-BO) and ca. 23 h (d-DPA-d-BO) at the luminance of 1000 cd m$^{-2}$, which represent about 20-fold increase when compared to that based on BN3 only, i.e. 1.1 h. To have a better understanding of these enhancements, we further investigate the photolysis of the solid-state thin films of BN3 (10%) doped in mCBP, and BN3 (10%) with DPA-BO (15%) doped in mCBP (Supplementary Fig. 46). We observe an enhanced photostability in the presence of DPA-BO, with the reduced drop in the PL intensity from 62.0% (BN3) to 46.6% (BN3 with DPA-BO), as well as a reduced delayed lifetime (from 224.2 μs to 196.3 μs), possibly attributed to the accelerated RISC process. Finally, we also attempted to fabricate triplet-triplet upconversion (TTU) OLEDs based on this series of TADF compounds. The discussion of the EL characteristics for these OLEDs is included in Supplementary Note 1. The spatial localization of 3 wt% DPA-BO doped in 9-(1-naphthalenyl)−10-(4-(2-naphthalenyl)phenyl)anthracene (BH) is shown in Supplementary Fig. 47, while the EL characteristics are illustrated in Supplementary Fig. 48−50 and summarized in Supplementary Table 12−14; and the key performance of recently reported blue OLEDs with CIE$y$ ≤ 0.1 is listed in Supplementary Table 15. Considering the operational stability of the devices, similar observations have been found in the TTU OLEDs, demonstrating the versatility of the current approach to be applied to different types of devices.

## Discussion

In conclusion, we present the subatomic modification of TADF emitters with targeted deuteration of donor and/or acceptor units and study the deuterated effect on their photophysical behavior, intrinsic stability and device operational stability. We found that the photostability of the deuterated isotopic analogues increases with the degree of deuteration, attributed to the higher intrinsic stability of the molecules with the increased number of deuterated substitutions through an additive effect of the multiple C−D bonds. We demonstrate that the deuteration of acceptor, in addition to a deuterated donor, is critical to boost the device lifetime of LT$_{90}$ from 15 h (deuterated donor only) to 23 h (deuterated donor and acceptor), an appreciable enhancement at the luminance of 1000 cd m$^{-2}$. Given the lack of sufficient attention on the effect of deuterated acceptors on enhancing the molecular stability across the field, we envisage that our present work provides insights and demonstrates the importance on modification of the chemical structures of materials at a subatomic level towards prolonging the device operational stability. Building upon our current study, we anticipate that such strategy could be applied to highly emissive narrowband multi-resonance TADF systems to fully harness the potential of these promising candidates for OLED applications.

## Methods
### Materials
2,5-dibromo-1,3-difluorobenzene, potassium carbonate, sodium tert-butoxide and tris(dibenzylideneacetone)dipalladium were purchased from Leyan. Phenol, Aniline-$d_5$ and bromobenzene-$d_5$ were purchased from Aladdin. Phenol-$d_5$, diphenylamine and boron tribromide were purchased from Sigma-Aldrich. Tri-tert-butylphosphine was purchased from Adamas. All solvents were purchased from General-Reagent. All commercially purchased chemicals were used without any further purification.

### Characterization of the TADF Compounds
$^{1}$H and $^{13}$C{$^{1}$H} NMR spectra were recorded by Bruker Avance 400 (400 MHz for $^{1}$H or 101 MHz for $^{13}$C{$^{1}$H} nuclei) Fourier-transform

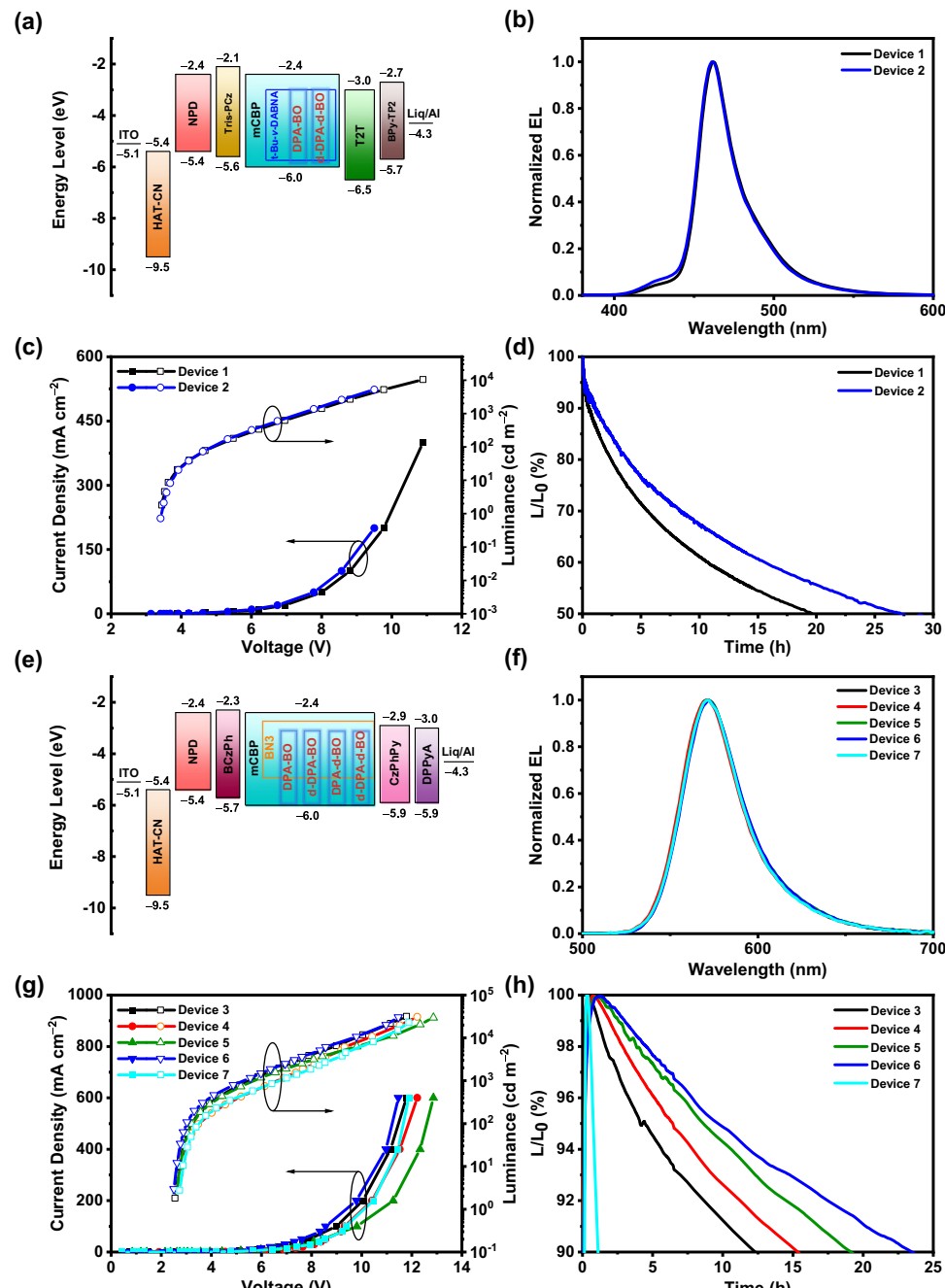

**Fig. 6 | Device characteristics of the vacuum-deposited TADF-sensitized OLEDs.**
**a**, **e** Device configuration and energy-level diagrams. **b**, **f** Normalized EL spectra.
**c**, **g** $J$–$V$–$L$ plots. **d**, **h** Operational lifetime projected at 1000 cd m$^{-2}$. Device 1 - t-Bu-$v$-
DABNA (1 wt%) + DPA-BO (15 wt%), Device 2 - t-Bu-v-DABNA (1 wt%) + d-DPA-d-BO

(15 wt%), Device 3 - BN3 (10 wt%) + DPA-BO (15 wt%), Device 4 - BN3 (10 wt%) + d-
DPA-BO (15 wt%), Device 5 - BN3 (10 wt%) + DPA-d-BO (15 wt%), Device 6 - BN3 (10 wt
%) + d-DPA-d-BO (15 wt%), Device 7 - BN3 (10 wt%). All devices were doped in mCBP.

NMR spectrometer with chemical shifts reported relative to tetra-
methylsilane (δ = 0 ppm) in chloroform and dimethyl sulfoxide. High
resolution electrospray ionization (ESI) mass spectra were recorded
by Thermo Scientific Exactive Benchtop LC/MS Orbitrap Mass
Spectrometer. FT-IR spectra were measured by infrared spectro-
photometer (Thermo Scientific Nicolet iS 50) from 4000 cm$^{-1}$ to
400 cm$^{-1}$.

### Absorption, emission spectra and PLQY measurements
The UV-vis absorption spectra were recorded by Cary 500 UV−vis
(Agilent Technology) spectrophotometer equipped with a xenon flash
lamp. Steady-state excitation and emission spectra were recorded by

Edinburgh Instruments Ltd FS5 spectrofluorometer. Delayed emission
spectra were recorded by Edinburgh Instruments Ltd LP 980 transient
absorption spectrometer. Liquid nitrogen was placed into Optistat DN
(Oxford Instruments) for temperature-dependent measurements
(77−298 K). Solid-state photophysical measurements were also per-
formed for vacuum-deposited thin films inside Optistat DN. Excited-
state lifetimes of solution and thin film of samples were measured in a
conventional laser system. The excitation source was EPL-375 from
Edinburgh Instruments which is a laser with wavelength ranging from
369 to 381 nm and output power (pulsed) of 0.11 to 0.15 mW. Lumi-
nescence decay signals were recorded by FLS 980 and analyzed by
exponential fitting model. Absolute PLQYs in thin films were measured

**Table 3 | Device characteristics of the vacuum-deposited TADF-sensitized OLEDs**

| Device | $V_{on}$[a] (V) | $L_{max}$[b] (cd m$^{-2}$) | $\lambda_{max}$[c] (nm) | CIE[d] (x, y) | EQE$_{max/100/1000}$[e] (%) | $\Delta_{roll-off}$[f] (%) | CE[g] (cd A$^{-1}$) | PE[h] (lm W$^{-1}$) | LT$_{90}$[i] (h) | LT$_{50}$[i] (h) |
|---|---|---|---|---|---|---|---|---|---|---|
| 1 | 3.4 | 10399 | 462 | 0.13, 0.10 | 5.1/4.2/3.4 | 33 | 4.6 | 4.1 | - | 19.7 |
| 2 | 3.4 | 10689 | 461 | 0.13, 0.09 | 5.0/4.2/3.4 | 32 | 4.3 | 3.8 | - | 27.3 |
| 3 | 2.5 | 31763 | 572 | 0.50, 0.50 | 19.5/14.2/3.8 | 81 | 70.4 | 80.2 | 12.3 | - |
| 4 | 2.6 | 30854 | 572 | 0.49, 0.50 | 20.9/12.8/3.4 | 84 | 75.1 | 82.9 | 15.4 | - |
| 5 | 2.6 | 29697 | 571 | 0.50, 0.50 | 16.5/13.1/3.5 | 79 | 59.9 | 64.8 | 19.1 | - |
| 6 | 2.5 | 30426 | 572 | 0.50, 0.50 | 18.5/15.1/3.6 | 81 | 67.7 | 76.9 | 23.4 | - |
| 7 | 2.7 | 22658 | 572 | 0.50, 0.50 | 15.2/10.2/2.2 | 86 | 56.3 | 58.0 | 1.1 | - |

[a]$V_{on}$ represents turn-on voltage.
[b]$L_{max}$ represents maximum luminance.
[c]$\lambda_{max}$ represents peak maximum.
[d]CIE coordinates are taken at luminance of 100 cd m$^{-2}$.
[e]EQE$_{max}$ represents maximum external quantum efficiency.
[f]Efficiency roll-off measured by $\Delta_{roll-off} = 1 - (EQE_{1000}/EQE_{max})$.
[g]CE represents maximum current efficiency.
[h]PE represents maximum power efficiency.
[i]Operational lifetime projected at 1000 cd m$^{-2}$.

by Hamamatsu Quantaurus-QY Absolute PL quantum yield spectrometer C11347-11.

### Ultraviolet photoelectron spectroscopy
The UPS data were measured using X-ray photoelectron spectrometer (ThermoFischer, ESCALAB Xi$^+$) under the operational vacuum pressure of $8 \times 10^{-10}$ Pa and energy of power source of 21.22 eV (He1 excitation source). Operating voltage was 12.5 kV and the accumulated signal was obtained in 10 cycles.

### Thermal analysis measurements
Thermogravimetric analysis was performed by Mettler TGA2 thermogravimeter where the sample weight losses were measured from 25 °C to 100 °C (kept at 15 min) then to 800 °C at a rate of 10 °C/min under nitrogen. Differential scanning calorimetry was carried out in SmartLab (Rigaku) by heating the samples from 50 °C to 350 °C at a rate 10 °C/min, followed by cooling process at the same rate. The samples were heated in two different cycles.

### Photolysis measurements
$2 \times 10^{-5}$ M of each compound in toluene solution was added to a quartz cell with path length of 1.0 cm, whereas the compounds doped in mCBP thin film were prepared by vacuum evaporation system FS-450. They were photo-irradiated with broadband light by using a xenon lamp (300 W, CEL-PF300-T6). The UV-vis absorption and photoluminescence spectra were recorded during photolysis over time. The compounds in neat film were fabricated in vacuum evaporation system FS-450 and photo-irradiated with broadband light from 150 W ozone-free xenon arc lamp in Optistat DN.

### Electron paramagnetic resonance
EPR measurements were performed with a Bruker A300 in nitrogen condition at room temperature. 5 mg of samples were weighed into the paramagnetic tube tested with light source 355 nm laser power 3 W light source model AO-L-355 and using solid 1,1-diphenyl-2-picrylhydrazyl as internal standard. A total of 12 data points of each sample was acquired in every 5 min for 1 h and 12 data points of selected samples were acquired every 30 min for 5 h.

### Time-of-flight secondary ion mass spectrometry
The emissive layer was formed by co-depositing respective TADF compounds, DPEPO and BH at different concentrations, with the thickness of 200 nm. ToF-SIMS was carried out in Nano TOF 2 instrument (ULVAC-PHI, Japan) equipped with bismuth as the primary ion source as well as Ar/GCIB/Cs ion as sputter source. The ToF-SIMS data was acquired on an area of $100 \times 100$ μm$^2$ by using 30 keV Bi primary ion beam, followed by a 3 s/cycle sputter of a $400 \times 400$ μm$^2$ area using 3 keV Ar ion beams. The sputter rate is 0.3 nm/s (Ar$^+$) for SiO$_2$. The sample holder with samples was sealed in a container filled with Ar gas and transferred into the instrument immediately before testing.

### Computational details
DFT and TDDFT calculations were carried out with the Gaussian 16 suite of programs[41]. By DFT, the ground-state (S$_0$) geometries of DPA-BO, d-DPA-BO, DPA-d-BO and d-DPA-d-BO were fully optimized in toluene with the hybrid Perdew, Burke, and Ernzerhof (PBE0) functional[42–44], in conjugation with the polarization continuum model (PCM)[45–47]. TDDFT[48–50] calculations at the same level associated with PCM were then carried out on the optimized S$_0$ geometries to compute the singlet-singlet transitions in the electronic absorption spectra of the compounds. Simulated absorption spectra of the compounds were generated by Multiwfn 3.8[51]. In order to gain insights into the excited states involved in thermally activated delayed fluorescence, the geometries of the lowest-lying singlet excited state (S$_1$), lowest-lying triplet excited state (T$_1$) and second lowest-lying triplet excited state (T$_2$) of the compounds were optimized using TDDFT with the PBE0 functional[42–44]. On the basis of the optimized geometries, vibrational frequencies were calculated and all stationary points were verified to be minima on the potential energy surface, as there were no imaginary frequencies observed (NIMAG = 0). The Cartesian coordinates of the optimized ground- and excited-state geometries are given in Supplementary Table 12–21. For all calculations, the 6-31 G(d,p) basis set[52–54] was employed for all elements. A pruned (99,590) grid was used for numerical integration. Spin-orbit coupling (SOC) calculation was performed using the ORCA 5.0.3 program by quasi-degenerate perturbation theory from TDDFT[29]. SOC matrices of all the compounds between the singlets and triplets were calculated and printed with B3LYP/def2-TZVP method in gas state, from which the total spin-orbit coupling matrix elements ($<S|\hat{H}_{SOC}|T>$) were calculated[55,56]. On the basis of the optimized S$_0$ geometries, the zero-point energies of the compounds and their respective donors and acceptors moieties were calculated using Multiwfn 3.8[51], while the bond dissociation energies were calculated at the M06-2X/6-311 + + G(d,p) level[57]. Based on the optimized S$_0$ and S$_1$ geometries, the displacement ($\Delta Q$), Huang-Rhys factors ($S_j$) and reorganization energies ($\lambda_j$) of the normal modes of the compounds were calculated by the Molecular Materials Property Prediction Package (MOMAP)[58]. In order to investigate the intermolecular interactions in the solid-state thin film, geometry optimization of the dimer of DPA-BO has been performed at the M06/6-31 G(d,p) level.

## Device fabrication and characterization

DPEPO-based devices were prepared with the configuration of ITO/ HATCN (10 nm)/TAPC (40 nm)/TCTA (10 nm)/MCP (10 nm)/$x$ wt% TADF compounds:DPEPO (25 nm)/TPBi (40 nm)/Liq (1 nm)/Al (100 nm), in which HATCN, TAPC/TCTA, MCP, TPBi and Liq were used as hole-injection, hole-transporting, electron-blocking, electron-transporting and electron injection layers, respectively. The emissive layer was formed by co-depositing respective TADF compound and DPEPO at different concentrations. For TADF-sensitized OLEDs, devices were prepared with architectures of ITO/HATCN (10 nm)/NPD (40 nm)/Tris-PCz (10 nm)/1wt% t-Bu-$v$-DABNA:15 wt% TADF compounds:mCBP (25 nm)/T2T (10 nm)/BPy-TP2 (40 nm)/Liq (1 nm)/Al (100 nm) and ITO/ HATCN (5 nm)/NPD (30 nm)/BCzPh (10 nm)/10 wt% BN3:15 wt% TADF compounds:mCBP (25 nm)/CzPhPy (10 nm)/DPPyA (30 nm)/Liq (1 nm)/ Al (120 nm). For TTU OLEDs, devices were fabricated with the architecture of ITO/3 wt% p-dopant:HTL1 (10 nm)/HTL1 (50 nm)/HTL2 (5 nm)/emissive layer (20 nm)/ETL1 (5 nm)/50 wt% Liq:ETL2 (30 nm)/Liq (2 nm)/Al (120 nm), in which the emissive layer was built by co-depositing 3 wt% of respective TADF compound and BH host simultaneously. All OLEDs were fabricated on the pre-patterned ITO-coated glass substrates with sheet resistance of 15 Ω per square. The substrates were cleaned with Decon 90, rinsed with deionized water and ethanol, then dried in an oven, and finally treated in an ultraviolet ozone chamber for 20 min before loading into a vacuum evaporation system. The films were sequentially deposited at a rate of 0.1–0.2 nm s$^{-1}$ at a pressure of ca. $3 \times 10^{-6}$ Torr without vacuum break. A shadow mask was used to define the pixel size of 0.09 cm$^2$. The thickness and deposition rate were monitored in situ during deposition by an oscillating quartz thickness monitor. Current density–voltage–luminance characteristics and EL spectra were measured with a Keithley 2400 semiconductor characterization system. The operational lifetime of the OLED devices was measured by a PR–OLEDLT-16 OLED aging tester.

## Measurement of transient electroluminescence

Transient EL data were obtained using Tektronix AFG 1062 signal generator working under pulse mode, with 10 KHz repetition rate, 10% duty cycle, to excite the sample. The resulting EL signal was detected by Zolix OmniFluo990-H which integrated with sample chamber, monochromator, PMT (CR131, response time 2.2 ns), and signal acquisition device DSC900PC.

## Data availability

The data generated in this study are provided in the Supplementary Information. Source data are provided with this paper.

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

## Acknowledgements

The work described in this paper was fully supported by a grant from the National Natural Science Foundation of China (Project No. 22275114), Science and Technology Planning Project of Shenzhen Municipality (Grant number: WDZC20220817160017003; WDZC20220810152404001), Cross-Disciplinary Research Fund of Tsinghua Shenzhen International Graduate School (SIGS), Tsinghua University (Grant number: JC2022003), Overseas Research Cooperation Fund Research Plan of Tsinghua SIGS (Grant number: HW2023006) and Scientific Research Startup Fund of SIGS, Tsinghua University (Grant numbers: QD2021027C and QD2023001C). W.-L.C. acknowledges the receipt of postgraduate studentships from the Science and Technology Planning Project of Shenzhen Municipality. The computations were performed using the SIGS research computing facilities.

## Author contributions

M.-C.T. initiated and designed the research. M.-C.T. and S.J. designed the TADF compounds. M.-C.T., S.J., C.W., and X.S. conducted the synthesis, characterization, and photophysical measurements of the compounds. W.-L.C., W.L., and W.C. carried out the OLED fabrication and characterizations. S.L., M.W., M.N., and W.K.T. performed and analyzed the computational calculations. M.-C.T., G.W., S.S.C., and Q.S. supervised the work. All authors discussed the results and contributed to the manuscript.

## Competing interests

The authors declare no competing interests.
