## [Peer Review File · Nature Communications]

Enhancing Operational Stability of OLEDs Based on Subatomic Modified Thermally Activated Delayed Fluorescence CompoundsREVIEWER COMMENTS

Reviewer #1 (Remarks to the Author):

The study conducted by Tang et al. described an intriguing investigation into the efficacy of deuterated thermally activated delayed fluorescence (TADF) materials in enhancing the operational stability of organic light-emitting diodes (OLEDs). The authors comprehensively examine the effect of deuteration on the TADF properties of the compounds. Through independent deuteration of the donor and the acceptor units, the team investigated the correlation between isotopic effect and the corresponding device operational stability. In detail, a series of isotopic analogues were synthesized, namely DPA-BO, d-DPA-BO, DPA-d-BO, and d-DPA-d-BO.

The results found that the compound photostability increased with the degree of deuteration as confirmed by time-dependent UV-vis and PL spectra in the tests of solution, neat film and doped film. Theoretical calculations also supported an increased bond dissociation energy of the C-D bond compared to regular C-H bond was beneficial to the molecule stability. In particular, deuteration of the acceptor, in addition to a deuterated donor, significantly increased the device lifetime by 57% upon using as the photosensitized materials of the emissive layer in the OLED stack. (from 14 h to 22 h).

Overall, this manuscript is well-organized and well-written, which is also timely and pertinent to the field of OLEDs, and the approached developed in this study can benefit the readership of the related fields of deuterated TADF materials and OLEDs. With moderate revisions and supplementary data, I would recommend this manuscript for publication in Nature Communications. Specific comments are as follows.

Major issues:

1. It was claimed by the author that the OLED stability may be related to the strong C-D bond than that of the C-H bond. In order to further understand the behind reason, the reviewer strongly suggest the authors add additional information by the DFT calculation.
2. I would like to request authors to provide the photostability data for the case of the mCBP host and BN3-mCBP host systems. Then, we can surely compare the PL and EL stabilities with appropriate discussion.
3. At the end of the manuscript, the author suggests such strategy could be adopted into highly robust multi-resonance TADF systems. This sounds a bit abrupt. Please elaborate how could the approach in this system applied to MRTADF systems, and what could be the potential challenges?

Minor issues:

Page 4 Line 8: "...mechanism of such effect has yet been..." should be "...mechanism of such effect has not yet been..."

Page 4 Line 11: The author may consider revising "full potential of fully deuterated materials" to "capacity of fully deuterated materials".

Page 4 Line 19: “a series of deuterated isomers” They are not isomers, they are isotopologues. Please double check it throughout the entire manuscript.

Page 5 Line 27: “...simulated adsorption spectrum of all compounds” should be “...simulated adsorption spectra...”

Page 7 Line 7: activation energy (E_a), E should be italicized.

Page 7 Line 8: “is beneficial to suppress complexation reaction” Please consider revising to “facilitates suppression of complexation reaction”

Page 7 Line 11: “final compounds” are vague. Please change to “the corresponding end products”.

Page 11 Line 15: “g value” should be “g value”

Reviewer #2 (Remarks to the Author):

Tang and co-workers reported a comprehensive study on the subatomic modified thermally activated delayed fluorescence emitters and their operation stability of organic light-emitting diodes (OLEDs). It should be mentioned that some research works have been reported and shown the positive isotopic effect on the donor moieties of the compounds *Nat. Photon.* 16, 843–850 (2022) and *Chem. Eng. J.* 430, 132822 (2022), this work represents first demonstration on the deuteration of the acceptor units and its value to the OLED operational stability. The molecule structures are well-characterized and supported by NMR and high resolution MS. More importantly, the operational stability of OLEDs has been improved through the use of deuterated materials as both emitters and sensitizing materials, showing an improved OLED operational lifetime of up to 50%. After the authors address the following concerns, I would recommend this manuscript for publication in *Nature Communications*.

1. The delta E_{st} of these compounds in the solid-state thin films should be determined by PL transient emission spectra, instead of the steady state emission spectra as shown in the Figure S35.
2. The commercial source or the preparation of the deuterated chemicals should be provided.
3. Please double check the figure 3c with the table 1, the delayed lifetime seems to be inconsistent.
4. PL properties of the TADF emitter was studied in doped DPEPO film, but photostability test was carried out in doped mCBP film. Please provide the reason why different host materials were chosen.
5. Can the authors provided explanations for the limited device performances in non-sensitized devices and sensitized device 1 and device 2.
6. In Figure S38, why the intensity drop for DPA-BO is smaller than d-DPA-BO, please double check the data.

7. It is suggested to change light green lines to a darker color for clarity in the whole manuscript.

Reviewer #3 (Remarks to the Author):

Tang and co-workers reported a series of TADF emitters upon systematic deuteration. The authors clearly express their strategy for prolonging the device operational stability in the introduction section. They also investigated the effect of deuteration on photophysical behavior, intrinsic stability, and device operational stability. Overall, this manuscript is well-written and recommended for publication in Nature Communications after a minor revision as follows.

1. In the introduction section, on page 3, line 32, the authors mentioned that the increased PLQY of the near infra-red emitters is due to the lowering of the non-radiative decay rate (k_{nr}) and the reorganization energy (λ). Did the authors investigate the effect of deuteration on k_{nr} , λ and even the Huang-Rhys factor?
2. Besides deuterium, do the authors think of any isotope of other elements that can be applied in their future studies?
3. On page 7, line 5, regarding the statement “suggesting the need of a higher activation energy (E_a) for C–D bond breaking in the deuterated compounds”. In fact, the lowering of the ZPE can be observed in both the reactants and products, and this nearly cancels out for the BDE. However, the effect would be more pronounced for barriers as the transition states often have weak vibrations with low frequencies. As a result, investigating the transition state and the corresponding activation energy of the C(H), C(D) and C(N) bond dissociations would help further understand the reason for the OLED stability.
4. On page 25 of the supporting information regarding Figures S29 and S30. For the S1 state of DPA-BO, the LSOMO is on the DPA moiety, whereas the HSOMO is on the BO moiety. For the T1 state of DPA-BO, the LSOMO is on the BO moiety, while the HSOMO is on the DPA moiety. Could the authors please check if they are correct?
5. On page 7, line 17, the authors mentioned, “the proton abstraction reaction or high molecular adducts formation is probably less favourable”. Could the authors describe more specifically the possible pathways of the proton abstraction reaction and the subsequent degradation of the compounds?
6. On page 8, line 17, the authors mentioned “indicating the presence of strong intermolecular interactions in solid state.....”. What kind are those possible intermolecular interactions do the authors suggest?

Response to reviewers' comments

Reviewer #1 (Remarks to the Author):

The study conducted by Tang et al. described an intriguing investigation into the efficacy of deuterated thermally activated delayed fluorescence (TADF) materials in enhancing the operational stability of organic light-emitting diodes (OLEDs). The authors comprehensively examine the effect of deuteration on the TADF properties of the compounds. Through independent deuteration of the donor and the acceptor units, the team investigated the correlation between isotopic effect and the corresponding device operational stability. In detail, a series of isotopic analogues were synthesized, namely DPA-BO, d-DPA-BO, DPA-d-BO, and d-DPA-d-BO.

The results found that the compound photostability increased with the degree of deuteration as confirmed by time-dependent UV-vis and PL spectra in the tests of solution, neat film and doped film. Theoretical calculations also supported an increased bond dissociation energy of the C-D bond compared to regular C-H bond was beneficial to the molecule stability. In particular, deuteration of the acceptor, in addition to a deuterated donor, significantly increased the device lifetime by 57% upon using as the photosensitized materials of the emissive layer in the OLED stack. (from 14 h to 22 h).

Overall, this manuscript is well-organized and well-written, which is also timely and pertinent to the field of OLEDs, and the approached developed in this study can benefit the readership of the related fields of deuterated TADF materials and OLEDs. With moderate revisions and supplementary data, I would recommend this manuscript for publication in Nature Communications. Specific comments are as follows.

Response: We thank the reviewer for his/her positive feedback on our manuscript.

Major issues:

1. It was claimed by the author that the OLED stability may be related to the strong C-D bond than that of the C-H bond. In order to further understand the behind reason, the reviewer strongly suggest the authors add additional information by the DFT calculation.

Response: Thank you for your valuable comments on our manuscript. In response to your suggestion regarding the DFT calculation, we would like to kindly point out that we have indeed incorporated this calculation analysis in **Table S6**, which summarizes the bond dissociation energies (BDEs) of the C–H and C–D bonds as shown below. The BDE of the C–D bond is found

to be *ca.* 0.1 eV larger than that of the C–H bond upon the deuteration of both the DPA and BO moieties. Such energy difference is equivalent to *ca.* 2 kcal mol⁻¹, a value that is larger than the chemical accuracy (*i.e.* 1 kcal mol⁻¹), supporting that the C–D bond is significantly stronger than the C–H bond in the tested series of compounds in this study and that eventually enhancing their photo- and electroluminescence performances. While we have successfully computed the BDEs of the C–H and C–D bonds, we acknowledge the potential significance of the role of transition states in the C–H and C–D bond dissociation. In an attempt to optimize the transition states for the C–H (or C–D) bond dissociation, we encountered convergence issues, likewise, the typical case of H₂ dissociation, which highlights the limitations of density functional approximations. These limitations can be circumvented by performing calculations on the C–H or C–D bond activation reactions of the compounds by other molecules or radicals present in the system. Such calculations are undergoing in another TADF system and will be reported in due course.

Table S6. Bond dissociation energy (BDE) of the C–H or C–D bonds of the compounds optimized at the M06-2X/6-311++G(d,p) level.

Bond	BDE (eV)
C–H bond of DPA in DPA-BO	4.78
C–D bond of DPA in d-DPA-BO	4.87
C–H bond of BO in DPA-BO	4.86
C–D bond of BO in DPA-d-BO	4.95

2. I would like to request authors to provide the photostability data for the case of the mCBP host and BN3-mCBP host systems. Then, we can surely compare the PL and EL stabilities with appropriate discussion.

Response: As suggested by the reviewer, we have now obtained the photostability data of BN3 doped in mCBP, and BN3 with **DPA-BO** doped in mCBP, as well as their corresponding delayed photoluminescence (PL) decay in the solid-state thin films, illustrated in **Figure S46** as shown below. We observe an enhanced photostability in the presence of **DPA-BO**, with the reduced drop in the PL intensity from 62.0% (BN3) to 46.6% (BN3 with **DPA-BO**). These findings align with the results of our OLED operation test, as presented in **Table 3** that summarises the device characteristics of the vacuum-deposited TADF-sensitized OLEDs in the manuscript. Specifically, we observe an increased LT₉₀ from 1.1 h to 12.3 h in device 7 (BN3 doped in mCBP) and device 3 (BN3 with **DPA-BO** doped in mCBP), respectively, which were fabricated under the same device architecture but different composition in the emissive layer.

We attribute the enhanced stability of BN3 with **DPA-BO** doped in mCBP, as evidenced by both the PL and electroluminescence (EL) spectra, to the efficient Förster resonance energy transfer

(FRET) mechanism. The FRET process is facilitated by the higher singlet (3.01 eV) and triplet energies (2.93 eV) of the **DPA-BO** series compared to the singlet energy of BN3 (~2.20 eV). The FRET mechanism is illustrated in **Figure R1** as shown below. By introducing the TADF materials, **DPA-BO**, as assistant host or sensitizer for multi-resonance TADF BN3 guest, efficient FRET is achieved, as evidenced by the absence of emission leakage from **DPA-BO** (**Figure S46b** (PL) and **Figure 6f** (EL)).

Moreover, we observe a decrease in the excited state lifetime of BN3 from 224 μs to 196 μs upon the addition of **DPA-BO** (**Figure S46c**), indicating an efficient radiative decay from the S_1 state of BN3 followed by complete FRET. The reduction in the density of triplet excitons serves to suppress bimolecular triplet-triplet annihilation and triplet-polaron annihilation, which can otherwise lead to undesired exciton fusion processes, such as high-energy polaron formation, ultimately resulting in emitter degradation (*Adv. Opt. Mater.* **10**, 2200665 (2022)). As a result, the addition of **DPA-BO** into BN3 doped in mCBP exhibits enhanced PL and EL stabilities.

Figure S46. PL spectra of (a) **BN3** (10%) and (b) **BN3** (10%) + **DPA-BO** (15%) doped in mCBP over 60 min under irradiation of 300 W xenon lamp. (c) Delayed PL decay of **BN3** (10%) and **BN3** (10%) + **DPA-BO** (15%) doped in mCBP.

Figure R1. The illustration of FRET mechanism from **DPA-BO** to **BN3**.

Figure 6f. Device characteristics of the vacuum-deposited TADF-sensitized OLEDs. f, Normalized EL spectra.

Table 3. Device characteristics of the vacuum-deposited TADF-sensitized OLEDs.

Device	V_{on}^a (V)	L_{max}^b ($cd\ m^{-2}$)	λ_{max}^c (nm)	CIE^d (x, y)	$EQE_{max/100/1000}^e$ (%)	$\Delta_{roll-off}^f$ (%)	CE^g ($cd\ A^{-1}$)	PE^h ($lm\ W^{-1}$)	LT_{90}^i (h)	LT_{50}^i (h)
1	3.4	10399	462	0.13, 0.10	5.1/4.2/3.4	33	4.6	4.1	-	19.7
2	3.4	10689	461	0.13, 0.09	5.0/4.2/3.4	32	4.3	3.8	-	27.3
3	2.5	31763	572	0.50, 0.50	19.5/14.2/3.8	81	70.4	80.2	12.3	-
4	2.6	30854	572	0.49, 0.50	20.9/12.8/3.4	84	75.1	82.9	15.4	-
5	2.6	29697	571	0.50, 0.50	16.5/13.1/3.5	79	59.9	64.8	19.1	-
6	2.5	30426	572	0.50, 0.50	18.5/15.1/3.6	81	67.7	76.9	23.4	-
7	2.7	22658	572	0.50, 0.50	15.2/10.2/2.2	86	56.3	58.0	1.1	-

^{a)} V_{on} represents turn-on voltage.

^{b)} L_{max} represents maximum luminance.

^{c)} λ_{max} represents peak maximum.

^{d)} CIE coordinates are taken at luminance of $100\ cd\ m^{-2}$.

^{e)} EQE_{max} represents maximum external quantum efficiency.

^{f)} Efficiency roll-off measured by $\Delta_{roll-off} = 1 - (EQE_{1000}/EQE_{max})$.

^{g)} CE represents maximum current efficiency.

^{h)} PE represents maximum power efficiency.

ⁱ⁾ Operational lifetime projected at $1000\ cd\ m^{-2}$.

We have added the following discussion in the revised manuscript on page 13, lines 28–34:

“To have a better understanding of these enhancements, we further investigate the photolysis of the solid-state thin films of BN3 (10%) doped in mCBP, and BN3 (10%) with **DPA-BO** (15%)

doped in mCBP (**Figure S46**). We observe an enhanced photostability in the presence of **DPA-BO**, with the reduced drop in the PL intensity from 62.0% (BN3) to 46.6% (BN3 with **DPA-BO**), as well as a reduced delayed lifetime (from 224.2 μs to 196.3 μs), possibly attributed to the accelerated RISC process.”

3. At the end of the manuscript, the author suggests such strategy could be adopted into highly robust multi-resonance TADF systems. This sounds a bit abrupt. Please elaborate how could the approach in this system applied to MRTADF systems, and what could be the potential challenges?

Response: We appreciate the reviewer’s feedback regarding the suggestion made at the end of the manuscript. We agree that further elaboration is necessary to clarify how the approach employed in this system could be applied to MRTADF systems.

Narrowband emissive organoboron emitters featuring the multi-resonance (MR) effect have emerged as crucial components for constructing high-performance OLEDs with pure emission colours. These MR organoboron emitters exhibit high-efficiency narrowband TADF through triplet-to-singlet reverse intersystem crossing (RISC). **Figure R2** illustrates the molecular structures of DABNA-1 and v-DABNA as examples (*Comm. Chem.* 5, 149 (2022)). In recent reports, the synthesis procedures of these emitters have employed carbazole, diarylamine or phenol derivatives as building blocks. Our study demonstrates that these starting materials can be readily replaced with deuterated counterparts, opening possibilities for a new class of deuterated MRTADF emitters for OLED applications. Building upon our current study, it is reasonable to expect that the OLED operation stability of these deuterated emitters can be further enhanced. However, achieving meaningful improvements in stability requires precise control over deuteration and careful optimization. The challenges of experimental complexity and scale-up must also be addressed to enable practical implementation. These aspects call for focused efforts in deuteration control and optimization strategies to fully harness the potential of deuterated MRTADF emitters for OLED technologies.

Figure R2. Molecular structures of DABNA-1 and v-DABNA.

We have now amended the discussion for this part in the revised manuscript on page 17, lines 14–16:

“Building upon our current study, we anticipate that such strategy could be applied to highly emissive narrowband multi-resonance TADF systems to fully harness the potential of these promising candidates for OLED applications.”

Minor issues:

Page 4 Line 8: “...mechanism of such effect has yet been...” should be “...mechanism of such effect has not yet been...”

Page 4 Line 11: The author may consider revising “full potential of fully deuterated materials” to “capacity of fully deuterated materials”.

Page 4 Line 19: “a series of deuterated isomers” They are not isomers, they are isotopologues. Please double check it throughout the entire manuscript.

Page 5 Line 27: “...simulated adsorption spectrum of all compounds” should be “...simulated adsorption spectra...”

Page 7 Line 7: activation energy (E_a), E should be italicized.

Page 7 Line 8: “is beneficial to suppress complexation reaction” Please consider revising to “facilitates suppression of complexation reaction”

Page 7 Line 11: “final compounds” are vague. Please change to “the corresponding end products”.

Page 11 Line 15: “g value” should be “g value”

Response: We have now amended the text in the manuscript, as shown below.

Page 4, lines 7–9: “...the underlying mechanism of such effect has not yet been studied in correlation with the corresponding device operational stability.”

Page 4, line 11: “...suggesting the capacity of fully deuterated materials, ...”

Page 4, lines 17–18: “...to study such effect in a series of deuterated isotopic analogues, ...”

Page 5, line 2–3: “The series of deuterated isotopic analogues was synthesized by Buchwald–Hartwig coupling reaction, ...”

Page 6, Figure 2: “Theoretical investigation of the deuterated isotopologues.”

Page 18, lines 4–5: “We found that the photostability of the deuterated isotopic analogues increases with the degree of deuteration, ...”

Page 5, line 25: “Based on the simulated absorption spectra of all compounds, ...”

Page 7, line 7: “... (E_a)...”

Page 7, lines 21–22: “...the strengthening of multiple C–D bonds in the compounds is suggested to prevent most of the proton abstraction reaction and subsequent radical formations.”

Page 7, lines 7–8: “It should be noted that the addition of ZPE values of DPA and BO precursor compounds are not equal to the ZPE of the final compounds.”

Page 12, line 15: “...and a single spin signal with g value...”

Reviewer #2 (Remarks to the Author):

Tang and co-workers reported a comprehensive study on the subatomic modified thermally activated delayed fluorescence emitters and their operation stability of organic light-emitting diodes (OLEDs). It should be mentioned that some research works have been reported and shown the positive isotopic effect on the donor moieties of the compounds Nat. Photon. 16, 843–850 (2022) and Chem. Eng. J. 430, 132822 (2022), this work represents first demonstration on the deuteration of the acceptor units and its value to the OLED operational stability. The molecule structures are well-characterized and supported by NMR and high resolution MS. More importantly, the operational stability of OLEDs has been improved through the use of deuterated materials as both emitters and sensitizing materials, showing an improved OLED operational lifetime of up to 50%. After the authors address the following concerns, I would recommend this manuscript for publication in Nature Communications.

Response: We appreciate the reviewer for the recognition of our work and the valuable comments.

1. The delta Est of these compounds in the solid-state thin films should be determined by PL transient emission spectra, instead of the steady state emission spectra as shown in the Figure S35.

Response: As suggested by the reviewer, we have determined the ΔE_{ST} of the compounds in the solid-state thin films using PL transient emission spectra shown in **Figure S38**, which also includes the steady-state emission spectra. In the 10% DPEPO thin film at 300 K and 77 K, the **DPA-BO** compounds exhibit a broad, structureless fluorescence and phosphorescence emission band, with emission maxima observed at 450 nm and 454 nm, respectively. To ensure accuracy and eliminate potential errors arising from emission originating from the S_1 state, emission spectra for all compounds were also obtained with a gate delay time of 100 μ s, and no significant changes in the emission maxima were recorded. We then determined the ΔE_{ST} of the compounds in the solid-state thin films by the emission maximum derived from the emission spectra obtained at 300 K and delayed 77 K. The estimated ΔE_{ST} values for all compounds are found to be approximately 0.08 eV, indicating an insignificant change of the ΔE_{ST} values determined either by the steady or

time-delayed emission spectra.

Figure S38. Normalized prompt and delayed emission spectra of 10 wt% (a) **DPA-BO**, (b) **d-DPA-BO**, (c) **DPA-d-BO** and (d) **d-DPA-d-BO** doped in DPEPO films at 300 K and 77 K.

We have now amended the discussion for this part in the revised manuscript on page 9, lines 1–4:

“the PL transient emission spectra recorded at low temperature and the steady-state emission spectra at room temperature, representing the phosphorescence and fluorescence emission of the compounds doped at 10 wt% in DPEPO (**Figure S38**), where an estimated $\Delta E(S_1-T_1)$ of *ca.* 0.08 eV is found for all compounds.”

2. The commercial source or the preparation of the deuterated chemicals should be provided.

Response: Thank you for the reminder. We apologize for the missing information and have now provided the source of the commercially available deuterated chemicals in the “Methods” section in the main text on page 18, lines 1–7 as follows:

“**Materials.** 2,5-dibromo-1,3-difluorobenzene, potassium carbonate, sodium *tert*-butoxide and tris(dibenzylideneacetone)dipalladium were purchased from Leyan. Phenol, Aniline-*d*₅ and bromobenzene-*d*₅ were purchased from Aladdin. Phenol-*d*₅, diphenylamine and boron tribromide

were purchased from Sigma-Aldrich. Tri-*tert*-butylphosphine was purchased from Adamas. All solvents were purchased from General-Reagent. All commercially purchased chemicals were used without any further purification.”

3. Please double check the figure 3c with the table 1, the delayed lifetime seems to be inconsistent.

Response: We apologize for the confusion. We have rectified the data by remeasuring the emission lifetime multiple times, ensuring the accuracy in the updated version. The excited state lifetimes of the **DPA-BO** compounds are determined to be in the range of 57–64 μs . Upon further examination, we found that the outlined delayed lifetime of **d-DPA-BO** in comparison to other isotopic analogues was due to an inadvertent mistake (old **Figure 3c**, left). The results have been updated in the revised **Figure 3c** (right), which is now consistent with the data in **Table 1**.

Figure 3c. Prompt (inset) and delayed PL decay of 10 wt% doped films. Left) old version. Right) updated version.

4. PL properties of the TADF emitter was studied in doped DPEPO film, but photostability test was carried out in doped mCBP film. Please provide the reason why different host materials were chosen.

Response: We apologize for the typo in the figure caption causing confusion to the reviewer (*i.e.* old Figure S40, captioned as “PL spectra of (a) **DPA-BO**, (b) **d-DPA-BO**, (c) **DPA-d-BO** and (d) **d-DPA-d-BO** in *neat films* over 5 h under irradiation of 150 W xenon lamp”). The photostability test was indeed carried out in doped DPEPO film for a fair comparison of the study in examining the PL properties of the TADF emitter.

We have now updated the figure caption of the corresponding results in **Figure S43** as “PL spectra of (a) **DPA-BO**, (b) **d-DPA-BO**, (c) **DPA-d-BO** and (d) **d-DPA-d-BO** doped in 10% DPEPO films over 5 h under irradiation of 150 W xenon lamp”.

On the other hand, we have also conducted the photolysis study in doped mCBP film (**Figures 4c** and **4d**), which allow us to compare the PL and operation stability tests of our sensitized OLED, where **DPA-BO**, BN3 and mCBP serve as the sensitizer for multi-resonance TADF, the guest material, and the host material, respectively. We observe a decrease in the excited state lifetime of BN3 from 224 μ s to 196 μ s (**Figure S46c**), indicating an efficient radiative decay from the S₁ state of BN3 followed by complete FRET. The reduction in the density of triplet excitons serves to suppress bimolecular triplet-triplet annihilation and triplet-polaron annihilation, which can otherwise lead to undesired exciton fusion processes, such as high-energy polaron formation, ultimately resulting in emitter degradation. These findings align with the results of our OLED operation test, as presented in **Table 3** that summarises the device characteristics of the vacuum-deposited TADF-sensitized OLEDs in the manuscript. It is observed that LT₉₀ increased from 1.1 h to 12.3 h in device 7 (BN3 doped in mCBP) and device 3 (BN3 with **DPA-BO** doped in mCBP), respectively, which were fabricated under the same device architecture but different composition in the emissive layer.

Figure S43. PL spectra of (a) **DPA-BO**, (b) **d-DPA-BO**, (c) **DPA-d-BO** and (d) **d-DPA-d-BO** doped in 10% DPEPO films over 5 h under irradiation of 150 W xenon lamp.

Figure 4. Photolysis results. UV-Vis spectra of **c**, DPA-BO, **d**, d-DPA-d-BO doped in mCBP over 10 min under irradiation of 300 W xenon lamp.

Figure S46. PL spectra of (c) Delayed PL decay of **BN3 (10%)** and **BN3 (10%) + DPA-BO (15%)** doped in mCBP.

Table 3. Device characteristics of the vacuum-deposited TADF-sensitized OLEDs.

Device	V_{on}^a (V)	L_{max}^b ($cd\ m^{-2}$)	λ_{max}^c (nm)	CIE^d (x, y)	$EQE_{max/100/1000}^e$ (%)	$\Delta_{roll-off}^f$ (%)	CE^g ($cd\ A^{-1}$)	PE^h ($lm\ W^{-1}$)	LT_{90}^i (h)	LT_{50}^i (h)
1	3.4	10399	462	0.13, 0.10	5.1/4.2/3.4	33	4.6	4.1	-	19.7
2	3.4	10689	461	0.13, 0.09	5.0/4.2/3.4	32	4.3	3.8	-	27.3
3	2.5	31763	572	0.50, 0.50	19.5/14.2/3.8	81	70.4	80.2	12.3	-
4	2.6	30854	572	0.49, 0.50	20.9/12.8/3.4	84	75.1	82.9	15.4	-
5	2.6	29697	571	0.50, 0.50	16.5/13.1/3.5	79	59.9	64.8	19.1	-
6	2.5	30426	572	0.50, 0.50	18.5/15.1/3.6	81	67.7	76.9	23.4	-
7	2.7	22658	572	0.50, 0.50	15.2/10.2/2.2	86	56.3	58.0	1.1	-

^{a)} V_{on} represents turn-on voltage.

^{b)} L_{max} represents maximum luminance.

^{c)} λ_{max} represents peak maximum.

^{d)} CIE coordinates are taken at luminance of $100\ cd\ m^{-2}$.

^{e)} EQE_{max} represents maximum external quantum efficiency.

^{f)} Efficiency roll-off measured by $\Delta_{roll-off} = 1 - (EQE_{1000}/EQE_{max})$.

^{g)} CE represents maximum current efficiency.

- h)* PE represents maximum power efficiency.
- i)* Operational lifetime projected at 1000 cd m⁻².

5. Can the authors provided explanations for the limited device performances in non-sensitized devices and sensitized device 1 and device 2.

Response: Thank you for the reviewer's question. As shown in the **Figure 6b**, the device characteristics of vacuum-deposited TADF-sensitized OLEDs with t-Bu-v-DABNA showed an emission band peaking at 461 nm, with CIE coordinates of (0.13, 0.09). According to a recent study (*Adv. Funct. Mater.* **32**, 2110356 (2022)), t-Bu-v-DABNA emits at 2.60 eV, with a delayed excited lifetime of 2.93 μ s and a k_{RISC} value of $2.54 \times 10^5 \text{ s}^{-1}$. This relatively long excited state lifetime could induce the formation of high-energy excitons through processes such as bimolecular triplet-triplet annihilation or triplet-polaron annihilation. These processes may contribute to the degradation of the emitters. It is worth noting that the OLED stack employed in our study may not be optimally suited for saturated blue OLED, for which the availability of suitable host material is limited. As a result, for the vacuum-deposited devices with **DPA-BO** sensitizing t-Bu-v-DABNA, there could be possible energy mismatch of the host materials. In addition, and high emission energy of t-Bu-v-DABNA could lead to the formation of high energy exciton ($> 5 \text{ eV}$), which is known to degrade the emissive layer rapidly and thus limiting the device operation lifetime of sensitized devices 1 and 2. As for non-sensitized device, the relatively long excited state lifetime of BN3 (224 μ s) is undesirable for achieving a reasonable device stability, which also causes the limited device performance.

6. In Figure S38, why the intensity drop for DPA-BO is smaller than d-DPA-BO, please double check the data.

Response: Thank you for pointing out this issue. To further validate the intensity drop in the UV-vis absorption spectra of the compounds, we repeated the measurements multiple times, maintaining consistent concentration of the solutions for all the samples to ensure accuracy in the measurements. We re-examined the UV-vis spectra of the compounds and incorporated the updated data in **Figure S41** of the revised manuscript. The revised figure illustrates the trends of UV-vis absorbance intensities for the series of compounds, which are in good agreement with their degree of deuteration.

Figure S41. UV-Vis spectra of (a) **DPA-BO**, (b) **d-DPA-BO**, (c) **DPA-d-BO** and (d) **d-DPA-d-BO** at 2×10^{-5} M in toluene solution over 40 min under irradiation of 300 W xenon lamp.

7. It is suggested to change light green lines to a darker color for clarity in the whole manuscript.

Response: As suggested, a darker green colour has been adopted in both the revised manuscript and the supplementary information for clarity.

Reviewer #3 (Remarks to the Author):

Tang and co-workers reported a series of TADF emitters upon systematic deuteration. The authors clearly express their strategy for prolonging the device operational stability in the introduction section. They also investigated the effect of deuteration on photophysical behavior, intrinsic stability, and device operational stability. Overall, this manuscript is well-written and recommended for publication in Nature Communications after a minor revision as follows.

Response: We sincerely thank the reviewer for his/her positive feedback on our manuscript.

1. In the introduction section, on page 3, line 32, the authors mentioned that the increased PLQY of the near infra-red emitters is due to the lowering of the non-radiative decay rate (k_{nr}) and the reorganization energy. Did the authors investigate the effect of deuteration on k_{nr} , and even the Huang-Rhys factor?

Response: We appreciate the insightful feedback provided by the reviewer, which prompted us to perform additional calculations on the k_r , k_{nr} and the Huang-Rhys factors (S_j). We found that there were no obvious trends in the displacement (ΔQ), S_j and reorganization energies (λ_j) with respect to the deuteration of the compounds, and the k_r ($1.18 \times 10^8 \text{ s}^{-1}$) and k_{nr} (*ca.* 1.4×10^{11}) remained mostly unchanged upon deuteration. The contribution of normal mode vibration to the reorganization energy is plotted in **Figure S32** and the selected normal modes are illustrated in **Figures S33** and **S34**. These results clearly demonstrate that the λ_j value is primarily influenced by low-frequency vibrational modes below 50 cm^{-1} , which correspond predominantly to the rotation of the phenyl rings in both the DPA and BO moieties.

We have now added the following discussion in the main text on page 7, lines 25–35, and page 8, lines 1–2:

“In addition, the effects of deuteration of the compounds on their Huang-Rhys factor (S_j), reorganization energy (λ_j), radiative (k_r) and nonradiative (k_{nr}) decay rate constants have been investigated. Shown in **Figure S32** are the plots of the computed λ_j as a function of normal mode wavenumbers for S_1 - S_0 . It is obvious that the λ_j is contributed predominantly by the low-frequency vibrational modes under 50 cm^{-1} . These low-frequency vibrational modes are illustrated in **Figures S33** and **S34**, and they are mainly the rotation of the phenyl rings in both the DPA and BO moieties, which contribute predominantly to the distortion of the S_1 geometry from the S_0 geometry.

The displacement (ΔQ), Huang-Rhys factors (S_j) and reorganization energies of selected normal modes of the compounds are summarized in **Tables S8** and **S9**, and the computed k_r and k_{nr} values are summarized in **Table S10**. There is no obvious trend in ΔQ , S_j and λ_j upon the deuteration of the compounds, and the k_r ($1.18 \times 10^8 \text{ s}^{-1}$) and k_{nr} (*ca.* $1.4 \times 10^{11} \text{ s}^{-1}$) are nearly the same upon deuteration. Therefore, deuteration does not have significant effect on k_r and k_{nr} in this case, resemblance to our photophysical studies.”

Table S8. Vibrational frequencies (ω_j), displacement (ΔQ), Huang-Rhys factors (S_j) and reorganization energies (λ_j) of selected normal modes for S_0 of the compounds in toluene.

DPA-BO				d-DPA-BO			
ω_j (cm ⁻¹)	ΔQ	S_j	λ_j (meV)	ω_j (cm ⁻¹)	ΔQ	S_j	λ_j (meV)
16.59	-1082.23	44.26	91.02	16.05	-1120.44	45.90	91.35
39.61	-471.75	20.08	98.61	38.75	-468.79	19.40	93.19
462.01	11.87	0.15	8.49	62.81	-86.23	1.06	8.29
704.51	-7.43	0.09	7.74	435.19	-18.12	0.33	17.56
1671.31	-2.85	0.03	6.40	1648.17	3.49	0.05	9.35
DPA-d-BO				d-DPA-d-BO			
ω_j (cm ⁻¹)	ΔQ	S_j	λ_j (meV)	ω_j (cm ⁻¹)	ΔQ	S_j	λ_j (meV)
16.31	1083.23	43.61	88.21	12.25	1086.04	32.93	50.02
37.72	503.86	21.82	102.04	32.96	-582.53	25.48	104.15
121.20	43.98	0.53	8.03	120.36	-46.74	0.60	8.94
455.68	13.30	0.18	10.39	433.65	17.43	0.30	16.14
696.23	-7.36	0.09	7.41	1645.92	3.76	0.05	10.80

Table S9. Vibrational frequencies (ω_j), displacement (ΔQ), Huang-Rhys factors (S_j) and reorganization energies (λ_j) of selected normal modes for S_1 of the compounds in toluene.

DPA-BO				d-DPA-BO			
ω_j (cm ⁻¹)	ΔQ	S_j	λ_j (meV)	ω_j (cm ⁻¹)	ΔQ	S_j	λ_j (meV)
11.77	-1062.75	30.28	44.19	11.42	-1093.10	31.10	44.05
41.77	-402.03	15.38	79.64	41.70	-408.05	15.82	81.78
69.88	199.27	6.32	54.77	65.79	-218.53	7.16	58.39
94.66	133.99	3.87	45.44	86.90	-136.34	3.68	39.65
250.63	36.98	0.78	24.26	368.94	22.31	0.42	19.14
DPA-d-BO				d-DPA-d-BO			
ω_j (cm ⁻¹)	ΔQ	S_j	λ_j (meV)	ω_j (cm ⁻¹)	ΔQ	S_j	λ_j (meV)
11.35	1070.12	29.61	41.66	10.56	1085.17	28.32	37.06
39.77	420.11	15.99	78.84	35.12	-455.35	16.59	72.25
69.77	201.09	6.43	55.59	65.90	-246.95	9.16	74.80
94.54	135.00	3.93	46.01	86.16	139.34	3.81	40.71
250.02	-36.44	0.76	23.44	368.86	-22.95	0.44	20.25

Table S10. The computed radiative (k_r) and nonradiative (k_{nr}) decay rate constants of the S_1 state.

Compound	k_r (s^{-1})	k_{nr} (s^{-1})
DPA-BO	1.18×10^8	1.43×10^{11}
d-DPA-BO	1.18×10^8	1.45×10^{11}
DPA-d-BO	1.18×10^8	1.44×10^{11}
d-DPA-d-BO	1.19×10^8	1.48×10^{11}

Figure S32. Plots of computed reorganization energies as a function of normal mode wavenumbers for S_1 - S_0 of (a) **DPA-BO**, (b) **d-DPA-BO**, (c) **DPA-d-BO** and (d) **d-DPA-d-BO**.

Figure S33. Illustration of selected normal modes contributing to large reorganization energies for the S_0 state.

Figure S34. Illustration of selected normal modes contributing to large reorganization energies for the S_1 state.

2. Besides deuterium, do the authors think of any isotope of other elements that can be applied in their future studies?

Response: In addition to deuterium, researchers can explore other stable isotopes of elements such as carbon-13, oxygen-18 and nitrogen-15 for future investigations in organic electronics and the related fields. These isotopes might be used to modify organic materials or molecules in diverse ways, enabling the enhancement of specific material properties and a deeper understanding of the corresponding behaviour. Building upon the present study, our investigation into new classes of TADF and MRTADF emitters with phenol-C13 and carbazole-N15 derivatives as starting materials is currently in good progress. These findings will be reported in due course.

3. On page 7, line 5, regarding the statement “suggesting the need of a higher activation energy (E_a) for C–D bond breaking in the deuterated compounds”. In fact, the lowering of the ZPE can be observed in both the reactants and products, and this nearly cancels out for the BDE. However, the effect would be more pronounced for barriers as the transition states often have weak vibrations with low frequencies. As a result, investigating the transition state and the corresponding activation energy of the C–H, C–D and C–N bond dissociations would help further understand the reason for the OLED stability.

Response: We appreciate the reviewer’s suggestion. We have now optimized the transition states for the C–N bond dissociation of both **DPA-BO** and **d-DPA-d-BO**. Subsequently, we calculated the corresponding activation enthalpies for these reactions. The obtained results have now been incorporated into the updated version of **Table S7**. The calculated activation enthalpies for the C–N bond dissociation are approximately 0.90 eV, and there is minimal difference observed between the deuterated and non-deuterated compounds. In Qiao and coworkers’ study (*Nat. Commun.* **14**, 3927 (2023)), the BDE at the T_1 state was found to be linearly correlated with logarithm of device lifetime. Nevertheless, it is suggested that the BDE is more sensitive to the molecular structure than to the deuteration. Therefore, it is plausible that the BDEs found in our study do not show significant difference upon deuteration. Though the deuteration of **DPA-BO** does not strengthen the C–N bond, nor does it influence the rate of C–N bond dissociation, the strengthening of multiple C–D bonds in the compounds is suggested to prohibit most of the proton abstraction reactions and subsequent radical formations.

Table S7. Enthalpies of activation corresponding to the transition state of the C–N bond dissociation of **DPA-BO** and **d-DPA-d-BO** on the triplet potential energy surface.

Compound	Enthalpy of activation (eV)
DPA-BO	0.903
d-DPA-d-BO	0.904

In an attempt to optimize the transition states for the C–H (or C–D) bond dissociation, we encountered convergence issues, likewise, the typical case of H₂ dissociation, which highlights the limitations of density functional approximations. These limitations can be circumvented by performing calculations on the C–H or C–D bond activation reactions of the compounds by other molecules or radicals present in the system. Such calculations are undergoing in another TADF system and will be reported in due course. We delete the inappropriate statement “suggesting the need of a higher activation energy (E_a) for C–D bond breaking in the deuterated compounds”. In addition, we rewrite the paragraph after “The ZPE of these compounds and DPA and BO precursor compounds are listed in Table S4, whereas the BDE of the C–N bond and the C–H (or C–D) bond are summarized in Tables S5 and S6.” to enhance the clarity.

We have now revised the manuscript to clarify this point on page 7, lines 5–22:

“Regarding the C–N bond dissociation, the ZPE decreases from 262.9 kcal mol⁻¹ to 226.1 kcal mol⁻¹ upon increasing the degree of deuteration in general (**Table S4**). It should be noted that the addition of ZPE values of DPA and BO precursor compounds are not equal to the ZPE of the final compounds. The BDE of the C–N bond at the S₀ state is found to drop slightly upon deuterating the compounds, whereas the values at the T₁ state do not show apparent changes and are *ca.* 0.66 eV. The transition states of the C–N bond dissociation of the T₁ state of **DPA-BO** and **d-DPA-d-BO** have been located and their enthalpies of activation are summarized in **Table S7**. The enthalpies of activation are *ca.* 0.90 eV and they show insignificant difference upon deuteration. Therefore, from both the thermodynamic and kinetic aspects, deuteration of **DPA-BO** does not strengthen the C–N bond, nor does it influence the rate of C–N bond dissociation. On the other hand, the BDE of the C–D bond is higher than that of the C–H bond by *ca.* 0.1 eV. With multiple C–D bonds existing in the compounds (i.e. 10 in **d-DPA-BO**, 8 in **DPA-d-BO**; 18 in **d-DPA-d-BO**), the additive effect should be appreciable, and hence the proton abstraction reaction is thermodynamically less favourable and subsequent formation of high molecular adducts is probably suppressed.^{12,32} In short, upon deuteration of **DPA-BO**, though the C–N bond is slightly weakened, the strengthening of multiple C–D bonds in the compounds is suggested to prevent most of the proton abstraction reaction and subsequent radical formations. With these calculated results, we anticipate that the higher degree of deuteration on both donor and acceptor units should exert a more prominent effect on the device stability.”

4. On page 25 of the supporting information regarding Figures S29 and S30. For the S1 state of DPA-BO, the LSOMO is on the DPA moiety, whereas the HSOMO is on the BO moiety. For the T1 state of DPA-BO, the LSOMO is on the BO moiety, while the HSOMO is on the DPA moiety. Could the authors please check if they are correct?

Response: We apologize for the misleading information in the figures. We have checked and confirmed that, for the T₁ state of **DPA-BO**, the LSOMO is localized on the DPA moiety whereas the HSOMO is localized on the BO moiety. **Figure S30** has been revised with the correct spatial plots of HSOMO and LSOMO as shown below.

Figure S30. Spatial plots (isovalue = 0.03) of selected molecular orbitals of **DPA-BO** at the optimized T₁ state geometry.

5. On page 7, line 17, the authors mentioned, “the proton abstraction reaction or high molecular adducts formation is probably less favourable”. Could the authors describe more specifically the possible pathways of the proton abstraction reaction and the subsequent degradation of the compounds?

Response: Thank you for your comments. One of the possible pathways of the proton abstraction reaction is shown in the following **Scheme R1**, where the C–N bond would undergo cleavage, resulting in the formation of DPA and BO radicals that subsequently initiate further reaction by attacking neighbouring **DPA-BO** molecules. According to the work by Lee and co-workers in 2016 (*Adv. Opt. Mater.* **4**, 1281 (2016)) and other relevant studies, the C–N bond is generally acknowledged as the weakest chemical bond within the backbone structure of host materials. The low BDE of the C–N bond can lead to reduced stability and shorter lifetime of blue TADF-OLED devices (*Nat. Commun.* **14**, 3927 (2023)). In the case of **DPA-BO**, the weak C–N bonds may be prone to breaking due to the high excitation energy and prolonged triplet-state lifetime. Consequently, the resulting DPA and BO radicals can engage in reactions such as proton abstraction with neighbouring **DPA-BO** molecules, leading to formation of additional radicals. When considering deuterated **DPA-BO** molecules, the abstraction of deuterium is expected to be less favourable due to the primary isotopic effect as reported in other organic reactions. Based on this understanding, we propose that the introduction of deuterium into the compounds would enhance their resistance to such reactions, thereby improving the stability of OLEDs, as demonstrated in our study.

More comprehensive experimental and computational investigations, including mass spectrometry analysis of the generated products, as well as mechanistic and kinetic studies of both deuterated and non-deuterated **DPA-BO**, are necessary to enhance our understanding of the underlying mechanisms, with respect to thermodynamic parameters and the rate coefficients of the C–N bond cleavage and the subsequent proton abstraction processes. As such investigations are ongoing and would be beyond scope of the current study, we have decided not to include the proposed mechanism in the revised manuscript or the supplementary information. Nevertheless, we are revising the relevant text in the main body of the manuscript to provide additional insights and discussion on the topic.

Scheme R1. Proposed reaction pathways of the diarylamine radical towards **DPA-BO** compound.

6. On page 8, line 17, the authors mentioned “indicating the presence of strong intermolecular interactions in solid state.....”. What kind are those possible intermolecular interactions do the authors suggest?

Response: We suggest that the intermolecular interactions to be the non-covalent interactions such as π - π interaction, which is known to cause a red shift in photoluminescence spectra, in particular in the solid-state thin film. The photoluminescence spectra of the tested TADF molecules in toluene and in doped films are shown in **Figure 3a** and **Figure S36**, respectively. **Figure 3a** shows no shift in emission energy, while the luminescence peak redshifts from 429 nm to 452 nm as the **DPA-BO** concentration increased from 5 to 20 wt% in DPEPO films (**Figure S36**). These results indicate the presence of π - π interactions assumed the ground state dipole of this series of compound is of similar to that of DPEPO. To further investigate the possible intermolecular interactions in the solid-state thin film, we have additionally performed DFT calculations on the dimer form of **DPA-BO** at the at the M06/6-31G(d,p) level. The distance between the two planes of the BO moieties is 3.26 Å, which is smaller than the separation between the π - π planes (*ca.* 3.35 Å) (**Figure R3** and **Table R1**). Therefore, this additional computational result further confirms the presence of π - π interactions in the solid-state thin film.

Figure R3. Ground-state geometry of the dimer of **DPA-BO** optimized at the M06 level.

Table R1. Cartesian coordinates of the optimized geometry of the dimer of **DPA-BO** at the M06 level.

1	B	-2.974248	0.554762	-0.322829	55	B	-2.011193	0.918488	3.118541
2	C	-0.916441	-0.888000	0.080943	56	C	-4.275334	2.040688	2.796197
3	C	-1.479330	0.352500	-0.248440	57	C	-3.524123	0.907874	3.137929
4	C	-0.577633	1.399285	-0.484101	58	C	-4.252813	-0.245707	3.460555
5	C	0.797150	1.234935	-0.437743	59	C	-5.636895	-0.289542	3.454743
6	C	1.316298	-0.028640	-0.115205	60	C	-6.345893	0.867708	3.105962
7	C	0.449173	-1.099050	0.151932	61	C	-5.660991	2.044337	2.771469
8	H	1.443997	2.081977	-0.644344	62	H	-6.148031	-1.221019	3.678310
9	H	0.823140	-2.072818	0.451590	63	H	-6.191656	2.952329	2.499161
10	C	-3.071646	-1.874649	0.353969	64	C	-2.303384	3.334355	2.442748
11	C	-3.706791	-3.053257	0.750514	65	C	-1.875077	4.607031	2.060213
12	C	-3.788592	-0.729388	-0.040924	66	C	-1.405708	2.294465	2.749625
13	C	-5.087760	-3.120609	0.734750	67	C	-0.522590	4.865526	1.953139
14	H	-3.091529	-3.891088	1.068434	68	H	-2.627602	5.356091	1.828249
15	C	-5.187812	-0.866930	-0.082047	69	C	-0.039665	2.620137	2.650578
16	C	-5.838809	-2.025988	0.298211	70	C	0.406930	3.866509	2.253551
17	H	-5.584938	-4.034731	1.050646	71	H	-0.185641	5.850365	1.637535
18	H	-6.925043	-2.080783	0.266260	72	H	1.473132	4.066235	2.169047
19	C	-2.360718	2.943939	-0.845699	73	C	-2.253845	-1.519919	3.730268
20	C	-3.393296	2.012947	-0.624504	74	C	-1.376465	-0.457861	3.435156
21	C	-2.614306	4.286827	-1.133873	75	C	-1.803998	-2.816080	3.989270
22	C	-4.703162	2.530531	-0.660888	76	C	-0.008516	-0.790489	3.398865
23	C	-3.918732	4.741605	-1.167070	77	C	-0.449689	-3.087084	3.944046
24	H	-1.767636	4.948734	-1.297825	78	H	-2.540527	-3.585504	4.206220
25	C	-4.975056	3.861379	-0.915740	79	C	0.458410	-2.068726	3.644850
26	H	-4.117713	5.789884	-1.378536	80	H	-0.095804	-4.097303	4.136395
27	H	-6.002300	4.218830	-0.913868	81	H	1.523450	-2.281652	3.588757
28	H	-5.777838	-0.027676	-0.441616	82	H	0.694559	1.859522	2.904390
29	H	-5.537389	1.868057	-0.444514	83	H	0.714021	-0.020605	3.138122
30	O	-1.028096	2.643479	-0.782358	84	O	-3.614876	-1.400051	3.777836
31	O	-1.706253	-1.948051	0.387911	85	O	-3.663824	3.207758	2.476534
32	N	2.701227	-0.226970	-0.048032	86	N	-7.750391	0.852157	3.076796
33	C	3.598264	0.852700	0.170803	87	C	-8.505949	0.048012	3.964313
34	C	3.420659	1.725725	1.245885	88	C	-8.147855	-0.064059	5.310941
35	C	4.695076	1.021118	-0.676128	89	C	-9.640345	-0.623017	3.499795
36	C	4.322064	2.761107	1.458658	90	C	-8.906698	-0.848091	6.169380
37	H	2.566983	1.584562	1.906225	91	H	-7.271538	0.466379	5.675660
38	C	5.600164	2.049716	-0.449550	92	C	-10.403279	-1.389891	4.370482
39	H	4.829396	0.334245	-1.509000	93	H	-9.916559	-0.535412	2.451231
40	C	5.416361	2.926721	0.614775	94	C	-10.039448	-1.511773	5.707578
41	H	4.177346	3.431883	2.302963	95	H	-8.616301	-0.927675	7.214392
42	H	6.450629	2.171476	-1.116084	96	H	-11.283861	-1.906161	3.995196
43	H	6.123924	3.733316	0.788632	97	H	-10.634619	-2.118028	6.385536
44	C	3.269438	-1.516990	-0.232477	98	C	-8.440814	1.661981	2.136721
45	C	4.217671	-1.988358	0.677199	99	C	-9.440246	2.540790	2.555015
46	C	2.915711	-2.304996	-1.329669	100	C	-8.125074	1.573122	0.780467
47	C	4.804716	-3.233101	0.489617	101	C	-10.105513	3.327365	1.622267
48	H	4.491125	-1.365106	1.526230	102	H	-9.685845	2.598641	3.613266
49	C	3.494809	-3.555666	-1.501111	103	C	-8.778336	2.375746	-0.145250
50	H	2.181493	-1.929598	-2.038739	104	H	-7.362132	0.860403	0.473902
51	C	4.443072	-4.023973	-0.596292	105	C	-9.772556	3.256329	0.272675
52	H	5.543124	-3.590532	1.203446	106	H	-10.881930	4.011871	1.955613
53	H	3.211678	-4.163074	-2.357460	107	H	-8.519526	2.299044	-1.199477
54	H	4.899224	-5.000352	-0.738333	108	H	-10.290258	3.880227	-0.451426

References

Xie, W., Peng, X., Li, M., Qiu, W., Li, W., Gu, Q., Jiao, Y., Chen, Z., Gan, Y., Liu, K. K., Su, S.-J., Blocking the Energy Loss of Dexter Energy Transfer in Hyperfluorescence OLEDs Via One-Step Phenyl-Fluorene Substitution of TADF Assistant Host. *Adv. Opt. Mater.* 2022, 10, 2200665.

Naveen, K.R., Yang, H.I., Kwon, J.H., Double Boron-embedded Multiresonant Thermally Activated Delayed Fluorescent Materials for Organic Light-emitting Diodes. *Commun. Chem.* 2022, 5, 149.

Naveen, K. R., Lee, H., Braveenth, R., Karthik, D., Yang, K. J., Hwang, S. J., Kwon, J. H., Achieving High Efficiency and Pure Blue Color in Hyperfluorescence Organic Light Emitting Diodes using Organo-Boron Based Emitters. *Adv. Funct. Mater.* 2022, 32, 2110356.

Oh, C. S., Choi, J. M., Lee, J. Y., Chemical Bond Stabilization and Exciton Management by CN Modified Host Material for Improved Efficiency and Lifetime in Blue Phosphorescent Organic Light-Emitting Diodes. *Adv. Opt. Mater.* 2016, 4, 1281.

Meng, Q. Y., Wang, R., Wang, Y. L., Guo, X. W., Liu, Y. Q., Wen, X. L., Yao, C. Y., Qiao, J., Longevity Gene Responsible for Robust Blue Organic Materials Employing Thermally Activated Delayed Fluorescence. *Nat. Commun.* 2023, 14, 3927.

REVIEWERS' COMMENTS

Reviewer #1 (Remarks to the Author):

The authors have addressed all the issues I concerned, therefore, I suggested the acceptance of this revised manuscript.

Reviewer #2 (Remarks to the Author):

The authors have addressed the raised issues properly and the quality of the manuscript meets the requirement of Nat. Commun. I recommend publication in this version.

Reviewer #3 (Remarks to the Author):

I think the authors have addressed all the questions raised. Thus, I would like to recommend this manuscript for publication in its current form.